# Simulating the mid-Holocene, Last Interglacial and mid-Pliocene climate with EC-Earth3-LR

Qiong Zhang[1], Ellen Berntell[1], Josefine Axelsson[1], Jie Chen[1,2], Zixuan Han[1], Wesley de Nooijer[1], Zhengyao Lu[3], Qiang Li[1], Qiang Zhang[1], Klaus Wyser[4], Shuting Yang[5]

[1]Department of Physical Geography, Stockholm University, Stockholm, 10691, Sweden
[2]College of Earth and Environmental Sciences, Lanzhou University, Lanzhou, 730000, China
[3]Department of Physical Geography and Ecosystem Science, Lund University, Lund, 22100, Sweden
[4]Rossby Centre, Swedish Metrological and Hydrological Institute, Norrköping, 60176, Sweden
[5]Danish Meteorological Institute, Copenhagen, 2100, Denmark

*Correspondence to*: Qiong Zhang (qiong.zhang@natgeo.su.se)

**Abstract.** As global warming is proceeding due to rising greenhouse gas concentrations, the Earth system moves towards climate states that challenge adaptation. Past earth system states are offering possible modelling systems for the global warming of the coming decades. These include the climate of the Mid-Pliocene (~3 Ma), the Last Interglacial (~129-116 ka), and the Mid-Holocene (~6 ka). The simulations for these warm past periods are the key experiments in the Paleoclimate Model

Intercomparison Project (PMIP) phase 4, contributing to the Coupled Model Intercomparison Project (CMIP6). Paleoclimate modelling has long been regarded as a robust out-of-sample test-bed of the climate models used to project future climate changes. Here, we document the model setup for PMIP4 experiments with EC-Earth3-LR and present the large-scale features from the simulations for the mid-Holocene, the Last Interglacial, and the mid-Pliocene. Using the pre-industrial climate as a reference state, we show global temperature changes, large scale Hadley circulation and Walker circulation, polar warming,

global monsoons, and the climate variability modes (ENSO, PDO, AMO). The EC-Earth3-LR simulates reasonable climate responses during past warm periods, as shown in the other PMIP4-CMIP6 model ensemble. The systematic comparison of these climate changes in three past warm periods in an individual model demonstrates the model's ability to capture the climate response under different climate forcings, providing potential implications for confidence in future projections with EC-Earth model.

## 25  1 Introduction

Looking back on Earth's history, the warm climate that we are now experiencing is no exception. Several warm periods offer possible geological analogues for the future: the mid-Pliocene Warm Period (3.264–3.025 million years ago), the Last Interglacial (129,000–116,000 thousand year ago), and the Mid-Holocene (6000 years ago). Mid-Pliocene is the most recent period with atmospheric $CO_2$ compared to the present (approximately 400 ppmv) (Pagani et al., 2010), with average annual

surface temperatures of roughly 1.8 °C to 3.6 °C warmer than pre-industrial temperatures, reduced ice sheet size and increased sea levels (Haywood et al., 2013). Global mean annual temperatures during the Last Interglacial were approximately 0.8 °C

(maximum 1.3 °C) warmer than pre-industrial temperatures (Fischer et al., 2018) and the amplified seasonality characterized the northern latitudes (Marcott et al., 2013). During the Mid-Holocene period, temperatures were 0.7 °C warmer than pre-industrial temperatures (Otto-Bliesner et al., 2017), with increased seasonal temperatures and strengthened Northern Hemisphere (NH) monsoon (Marcott et al., 2013). The simulations for these warm periods are the key experiments in PMIP4/CMIP6 (Kageyama et al., 2018). These past warm periods provide essential insights into the climate processes that act in the context of higher $CO_2$ or stronger insolation. The mid-Pliocene experiment is designed to elucidate the climate system's long-term response to a concentration of atmospheric $CO_2$ close to present at 400 ppm (Haywood et al., 2016). Simulations in the mid-Holocene and last interglacial periods provide an opportunity to examine the impact of two different radiative forcing changes on the climate when other forcings were relatively similar to those present (Otto-Bliesner et al., 2017). Although the simulations for these past warm periods are not the perfect analogue for future projections, each experiment has distinct differences in the climate system's forcing and initial condition. Available paleo records provide a means of assessing whether we are correctly capturing the climate response to different forcings in the models being used for future climate change projections. These simulations can be used to evaluate the response of ocean circulation, Arctic sea ice, climate variability patterns (e.g., El Niño–Southern Oscillation), the global hydrological cycle, and regional monsoon systems to elevated concentrations of atmospheric $CO_2$ or the redistribution of solar energy around the globe. The evaluation will be a direct out-of-sample test of state-of-the-art models' reliability to simulate climate change, particularly climate warming.

EC-Earth3-LR is one of the models that contribute to PMIP4/CMIP6. For an individual climate model, the PMIP4 experiments contribution provides an opportunity to test the climate response to different forcing and overview the past climate change in one model system. All the participating modelling groups follow the PMIP4 protocols described in Kageyama et al. (2018). The detailed justification of the experimental protocols for each period are described separately in specific PMIP4 experiment design papers: Otto-Bliesner et al. (2017) for the mid-Holocene (the *midHolocene*) and lig127ka (the *lig127k*) experiments, and Haywood et al. (2016) for the mid-Pliocene experiment (the *midPliocene*). These past periods are characterized by changes in boundary conditions such as greenhouse gas concentrations (e.g., the *midPliocene*), orbital parameters (e.g., the *midHolocene* and the *lig127k*). The large-scale features from the PMIP4 multi-model ensemble have been evaluated for respective PMIP4 experiment for the *midHolocene* (Brierley et al., 2020b), *lig127k* (Otto-Bliesner et al., 2020), and *midPliocene* (Haywood et al., 2020a).

In the present paper, we document the experiments setup and the model performance for the pre-industrial, the mid-Holocene, the Last Interglacial and the mid-Pliocene with EC-Earth3-LR. We expect that a systematic comparison across different past warm climates in one individual model can improve our understanding of global warming through different forcings and provide a more reliable scientific base to evaluate the future climate projections.

In the following presentation, we use the *piControl*, the *midHolocene*, the *lig127k* and the *midPliocene* to refer the official PMIP-CMIP6 experiment names (except for the midPliocene-eoi400 that we use a short version *midPliocene*). We use abbreviations PI, MH, LIG, and MPlio to refer the four experiments in the illustrations.

## 2 The EC-Earth model description

### 2.1 EC-Earth model components and the configuration

The EC-Earth is a fully coupled earth system model that integrates several state-of-art components in the climate system including atmosphere, ocean, sea ice, land and biosphere. It is developed by the European consortium of 27 research institutions to date and widely used in various studies on climate change (Hazeleger et al., 2010;Hazeleger et al., 2012). The atmospheric component of EC-Earth3 is the Integrated Forecasting System (IFS) model of the European Center for Medium-Range Weather Forecasting (ECMWF), with Cycle 36r4. The revised Terrain Surface Hydrology ECMWF Scheme for Surface Exchanges

over Land (H-TESSEL) model is used for land surface use (Balsamo et al., 2009) as an integrated part of IFS. The ocean component is the Nucleus for the European Modeling of the ocean (NEMO) (Madec, 2008). It uses a tripolar grid of poles over northern North America, Siberia and Antarctica. The NEMO includes the Louvain-la-Neuve Sea-ice model version 3 (LIM3) (Vancoppenolle et al., 2012), a dynamic thermodynamic sea-ice model with five ice thickness categories.

Atmospheric–land and ocean–sea–ice components are coupled through OASIS (Ocean, Atmosphere, Sea Ice, Soil) (Valcke and Morel, 2006;Craig et al., 2017). The re-mapping of runoff from the atmospheric grid points to the runoff areas on the ocean grid has been re-implemented to be independent of grid resolution. This was achieved by introducing an auxiliary model component and relying on the interpolation routines provided by the OASIS coupling. In EC-Earth3-LR configuration, the time step of IFS at T159L62 resolution is 60 minutes except for radiation that is solved only every 3 hours (but updated with

cloud cover information in between two full computations). The time step for NEMO at ORCA1L75 resolution is 45 minutes, we use the same time step for the ocean and for the sea-ice. The coupling between IFS and NEMO is 3 hours.

EC-Earth3 earth system model version also includes the atmospheric chemistry component TM5, the dynamic vegetation model LPJ-Guess and the ocean biogeochemistry component PISCES. The EC-Earth3 model contributes to CMIP6 in several

configurations by using different resolutions or including different components. A detailed description of the EC-Earth3 and its contribution to CMIP6 is documented in (Döscher, 2021). For example, regarding the resolution, the standard version EC-Earth3 with atmospheric resolution T255L91 (horizontal ~ 80 km) and ocean resolution ORCA1L75 (horizontal 1°) is used in most MIPs (e.g., CMIP, DCPP, LS3MIP, PAMIP, RFMIP, ScenarioMIP, VolMIP, CORDEX, DynVarMIP, SIMIP). The high-resolution model version with atmospheric resolution T511L91 (horizontal ~ 25km) and ocean resolution ORCA025L75

(horizontal 0.25°) is used for DCPP and HighResMIP decadal predictions.

The PMIP4 experiments require long spin-up and equilibrium simulations which are computationally demanding. We thus use model configurations without atmospheric chemistry and ocean biogeochemistry at low resolution, with atmospheric resolution T159L62 (horizontal ~ 125km) and ocean resolution ORCA1L75. For the *piControl*, the *midHolocene*, the *lig127k*, *lgm* and the *midPliocene*, we have used the EC-Earth3-LR, a configuration of EC-Earth3 with only atmosphere, ocean and sea ice components at Low-Resolution (LR). The coupled vegetation LPJ-GUESS is used for the *past1000* simulation with model configuration named EC-Earth3-veg-LR. Despite using the lowest resolution among the EC-Earth3 CMIP6 MIPs configuration, EC-Earth3-LR has a relatively high resolution than the other PMIP4 models (Brierley et al., 2020a).

## 2.2 Implemented physics that needed for paleo-simulations

Two critical physical processes, i.e., orbital forcing and physics over the ice-sheet, crucial for paleoclimate simulations, are implemented in the EC-Earth3. Below, we describe the implementation of these two physics in the EC-Earth3.

### 2.2.1 Orbital forcing

Previous versions of the EC-Earth treated the orbital forcing as constants for the present-day climate in IFS by following the International Astronomical Union's recommendations. These formulas are not valid for dates too far away from the January 1st 2000 (if more than one century). For the paleoclimate simulations, the orbital forcing is calculated in the atmosphere component IFS (Berger, 1978). According to the model time, the new orbital parameterization uses the Julian calendar to calculate the cosine of the solar zenith angle. The Julian calendar has two type of year: a standard year of 365 days and a leap year of 366 days. To follow the Gregorian calendar used in IFS, which contains leap year, the orbital parameter calculation follows a simple cycle of three normal years and one leap year, giving an average year of 365.25 days long. The annual and diurnal cycle of solar insolation is thus represented with a repeatable solar year of precisely 365.25 days and with a mean solar day of precisely 24 hours, respectively. The year must fall within the range of $1950 \pm 10^6$. In the EC-Earth model run script, there are two ways to set the orbital forcing, either to change the three orbital parameters to known values corresponding to the specific year (defaults are for 1950) or to specify a number for the year. To determine the year, the exact year number applies for AD year, e.g., for the year 1850 AD, we set "*ifs_orb_iyear = 1850*". For the year before AD, the distance to reference year 1950 is given. For example, the mid-Holocene, which is 6000 years before the present day, can be specified as - 4950 (obtained from -6000+1950). For the equilibrium simulations, the orbital forcing is fixed for each year, and we set "*ifs_orb_mode = fixed*".

### 2.2.2 Albedo parameterization of snow on ice-sheet

The ice sheet albedo calculation is highly parametrized in the earlier version of EC-Earth by taking a constant value for areas with thick perennial snow cover. This is an important reason why the Greenland ice sheet surface mass balance is poorly resolved in the model. The surface scheme assumes the ice-sheet as 10 meters of perennial snow, and it is in thermal contact

with the underlying soil. The albedo and snow density are fixed at 0.8 and 300 kg m$^3$, respectively. The fixed, high snow albedo value over the ice sheet makes it difficult for the snow melting due to lack of albedo feedback. This process becomes crucial, especially for past periods with large ice-sheet extents, such as Last Glacial Maximum. Therefore, we introduce a new albedo parameterization of snow on the ice sheet, where snow melting becomes possible and occurs often. This varying snow albedo scheme is also active for the configuration of EC-Earth3 coupled with an interactive ice sheet model for the Greenland ice sheet, i.e., EC-Earth3-GrIS (Döscher, 2021). A comparison among different albedo parameterizations on the Greenland ice sheet is summarised in Helsen et al. (2017) using EC-Earth3. The time-varying snow albedo better represents the ice/snow albedo feedback and is beneficial for both cold and warm climates. From EC-Earth3, the snow is not allowed to accumulate for more than 9 meters of water equivalent. All excessive snow falls on the ice sheet is immediately redistributed into the ocean and will not accumulate on the ice sheet. This treatment has an advantage of energy and mass conserving, and the calving ice cools the ocean along the coast, which reduces the warm bias in the Southern Ocean.

The above mentioned two physics are included in the CMIP6 version EC-Earth3, with on and off options. For the PMIP4 experiments, we have applied these new physical implementations. However, other CMIP6 experiments, e.g., future projections have not used these new implementations.

## 3 The experiment design and setup with EC-Earth3-LR

### 3.1 PMIP4-CMIP6 protocols on climate forcings and boundary conditions

We follow the PMIP4 protocol and implement the climate forcings and boundary conditions for each PMIP4 experiment as documented in (Kageyama et al., 2018). The climate forcings implemented in the four experiments are listed in Table 1.

**Table 1**. PMIP4-CMIP6 protocols on forcings and boundary conditions.

|  | piControl | midHolocene | lig127k | midPliocene |
|---|---|---|---|---|
| Eccentricity | 0.016764 | 0.018682 | 0.039378 | Same as piControl |
| Obliquity (degrees) | 23.549 | 24.105 | 24.040 | Same as piControl |
| Perihelion - 180 | 100.33 | 0.87 | 275.41 | Same as piControl |
| $CO_2$ (ppm) | 284.3 | 264.4 | 275 | 400 |
| $CH_4$ (ppb) | 808.2 | 597 | 685 | Same as piControl |

| | | | | |
|---|---|---|---|---|
| N$_2$O (ppb) | 273.0 | 262 | 255 | Same as *piControl* |
| Topography & land-sea mask | Modern | Same as *piControl* | Same as *piControl* | Enhanced topography as in Haywood et al. (2016) |
| Ice sheets | Modern | Same as *piControl* | Same as *piControl* | As in Haywood et al. (2016) |
| Vegetation | CMIP DECK *piControl* | Same as *piControl* | Same as *piControl* | Salzmann et al. (2008) |

For all the four experiments, the solar constant is 1360.747 W m$^{-2}$, fixed for *piControl* at the mean value for the first two solar cycles of the historical period 1850-1871 (Eyring et al., 2016). The Vernal equinox is fixed at noon on March 21. Aerosols, including dust and volcanic, are set as the *piControl* value. Datasets for *midPliocene* from the PRISM4 reconstructions are downloaded from PlioMIP2 website: (https://geology.er.usgs.gov/egpsc/prism /7.2_pliomip2_data.html).

In agreement with the PMIP4 protocol, vegetation is prescribed as present-day climatology that is constant in time. The present-day climatological vegetation, based on ECMWF ERA-Interim (Dee et al., 2011), and specified as albedos and leaf area index (LAI) of the Moderate Resolution Imaging Spectroradiometer (MODIS) is used in *piControl*, *midHolocene* and *lig127k* simulations.

### 3.2 Boundary condition implementation for the *midHolocene* and *lig127k*

In the PMIP4 Tier 1 for the *midHolocene* and the *lig127k* experiments, the changes in boundary conditions are orbital parameters and greenhouse gas concentrations (Table 1). As described earlier, the setting for orbital forcing and greenhouse gas in the EC-Earth3-LR is straightforward. In the model run script, we set "*Orb_year = - 4950*" for the *midHolocene* and "*Orb_year = - 125050*" for the *lig127k*, the computed orbital parameters are the same as those listed in Table 1.

The changes in orbital parameters, particularly the higher obliquity in the *midHolocene* and *lig127k*, lead to large changes in the insolation's seasonal and latitudinal distribution, with stronger changes in the *lig127k* than in the *midHolocene* (Fig. 1). The global change in annual mean insolation is negligible. The Northern Hemisphere (NH) receives 20 W m$^{-2}$ more insolation during boreal summer June-August in the *midHolocene* and 48 W m$^{-2}$ in May-July in *lig127k*. While in boreal wintertime it receives 14 W m$^{-2}$ less in February-March in the *midHolocene* and 36 W m$^{-2}$ less during October-November in the *lig127k*. The maximum changes occur in the high latitudes, i.e., the Arctic region (60 - 90° N) receives 31 W m$^{-2}$ more in July in the *midHolocene* and 70 W m$^{-2}$ more in June in *lig127k*. The Southern Hemisphere (SH) receives 23 W m$^{-2}$ and 56 W m$^{-2}$ more in September and October in the *midHolocene* and *lig127k*. These large insolation changes in high latitudes cause polar

amplification (Sect. 4.4). The positive insolation anomalies in summer months lead to strong land-sea contrast and enhance the monsoons (Sect. 4.5).

The greenhouse gas concentration is provided by CMIP6, for the *piControl* the values are taken from the values at the year
1850 in the historical forcing files. An easy way for other PMIP4 periods is to create a new file by changing those values in year 1850 to the values specified in Table 1.

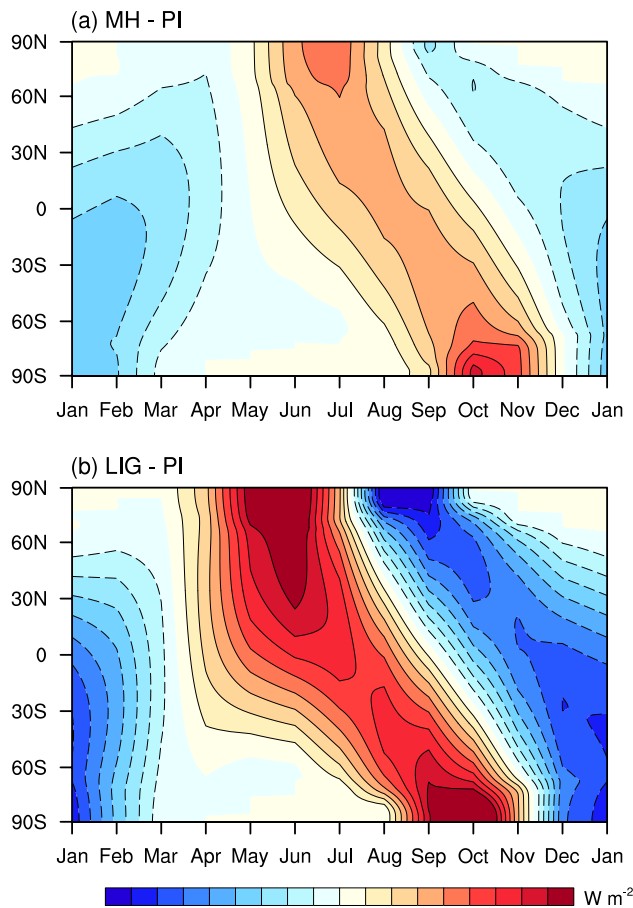

**Figure 1.** Changes in the latitudinal and seasonal distribution of insolation (W m$^{-2}$) as compared to the *piControl*, (a) the
*midHolocene*, (b) the *lig127k*.  The modern calendar is used, with the vernal equinox on March 21 at noon.

**3.3 Boundary condition implementation for the *midPliocene***

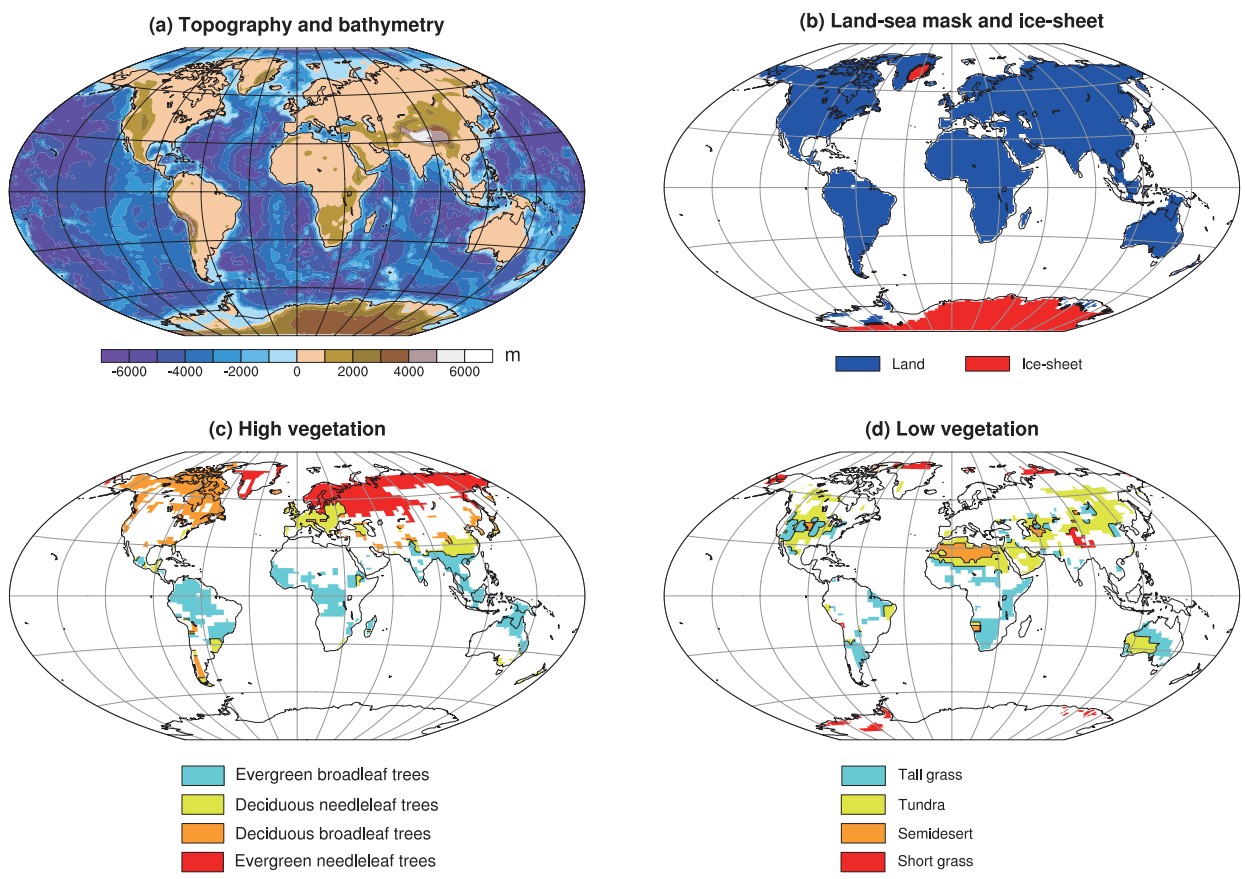

**Figure 2.** Prescribed boundary conditions for mid-Pliocene adapted to EC-Earth resolution. (a) topography and bathymetry (unit, metre), (b) land-sea mask and ice sheet, (c) high-vegetation and (d) low vegetation.

In the PlioMIP2 protocol, the mid-Pliocene period is centred at an interglacial peak (marine isotope stage KM5c) at around 3.025 Ma, which has similar orbital forcing as present-day. Since the differences between using the 3.025 Ma orbit and the present-day orbit are minimal (de Boer et al., 2017), the orbital parameters are kept the same as in *piControl*. For the PlioMIP2 Tier 1 experiment, the $CO_2$ concentration in the atmosphere is set at 400 ppm. $CH_4$ and $N_2O$ are specified as identical to those

190 in the *piControl* experiment. We have used the enhanced boundary condition according to PRISM4 reconstruction (Dowsett et al., 2016), which considers the change in dynamic topography associated with mantle flow and glacial isostatic adjustment due to Piacenzian ice loading. The topography changes include the closure of the Bering Strait and the straits through the Canadian Arctic Archipelago. We implement the topography, land-sea-mask, bathymetry, and ice sheet from PRISM4 to the horizontal grid in EC-Earth3-LR atmosphere and ocean (Fig. 2). The global distributions of soil and biome are modified to

match the *midPliocene* land-sea mask and ice-sheet reconstruction. Due to the change of land-sea mask, there is an emergence of new land areas, and rivers are routed to the nearest ocean point.

Following the PlioMIP2 protocol, we prescribe the vegetation from PRISM3 reconstruction data (Salzmann et al., 2008). The reconstructed vegetation is a 9-type mega-biome map classified after (Harrison and Prentice, 2003). In the EC-Earth3 land model HTESSEL, the vegetation type classification is based on the Biosphere-Atmosphere Transfer Scheme (BATS) model (Yang and Dickinson, 1996), which contains 20 biome classifications. Each grid point is characterized by a maximum of two dominant vegetation types, high vegetation and low vegetation, and their area fractions. The sum of the high and low vegetation coverage is always between 0 and 1. Therefore a translation from mega biome to HTESSEL classification is performed. We first re-gridded the mega-biome reconstruction data to the EC-Earth3 TL159 horizontal grid. The translation is then done grid by grid with the correspondence between the two biomes (Table. 2). We show the final vegetation types as four low vegetation types and four high vegetation of high vegetation in HTESSEL (Fig. 2). Due to the present translation scheme, each grid will have only one vegetation type, representing either high or low vegetation. See Table. 2 for details.

**Table 2**. The translation/correspondence between reconstructed mega-biome to EC-Earth HTESSEL vegetation type. H/L refer to the distinction between high (H) and low (L) vegetation.

| Reconstructed Mega biome | HTESSEL vegetation type (H/L) |
|---|---|
| Tropical Forest | Evergreen broadleaf trees (H) |
| Warm-Temperate Forest | Evergreen needleleaf trees (H) |
| Savanna/dry woodland | Tall grass (L) |
| Grassland/dry shrub | Short grass (L) |
| Desert | Semidesert (L) |
| Temperate forest | Deciduous broadleaf trees (H) |
| Boreal forest | Deciduous needleleaf trees (H) |
| Tundra | Tundra (L) |
| Dry tundra | Tundra (L) |

## 3.4 Initial conditions, spin-up and production

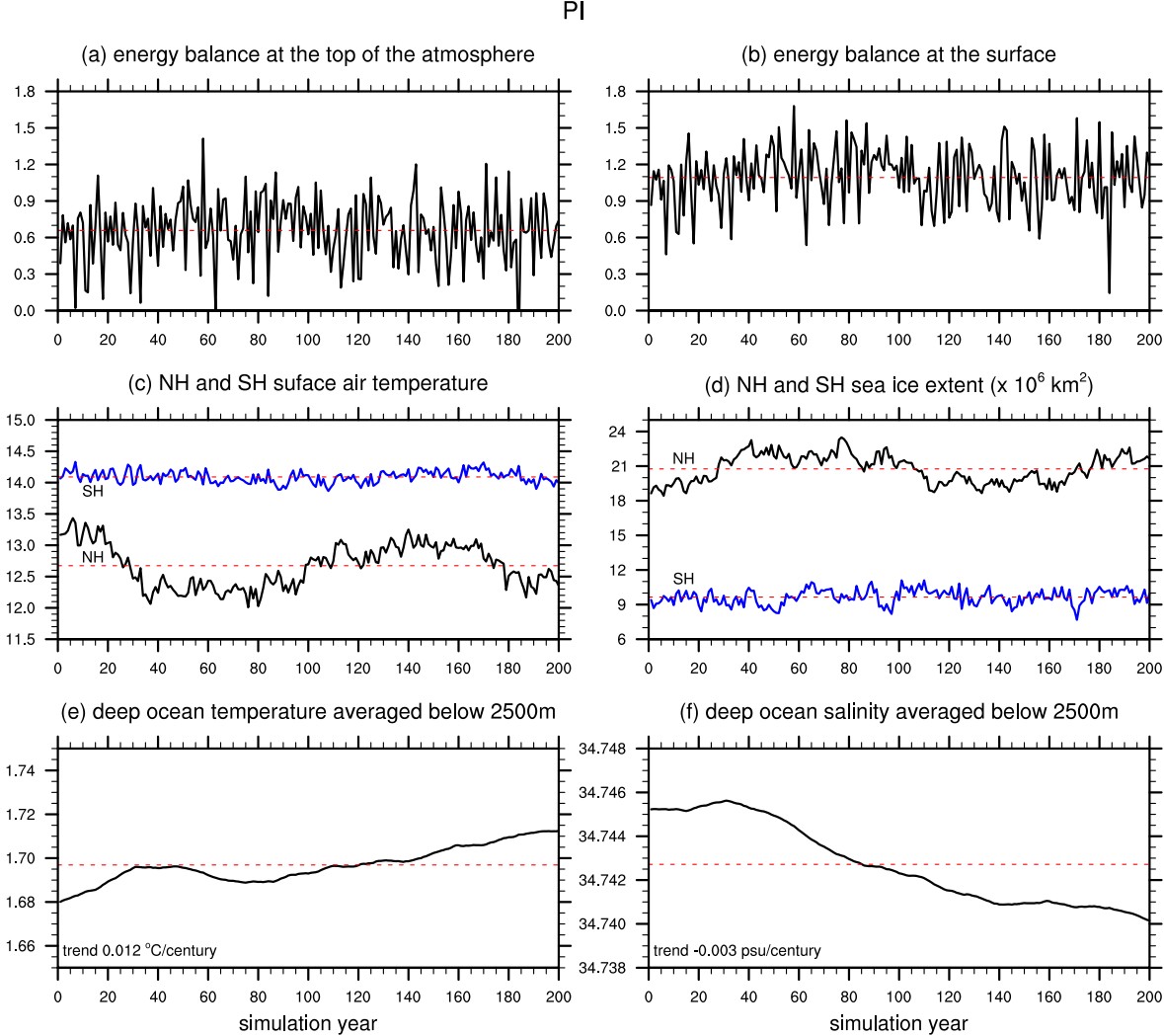

**Figure 3.** The evolution during the 200 years spin-up period for the *piControl* simulation using annual mean data. (a) global mean energy balance at the top of the atmosphere (W m$^{-2}$), (b) global mean energy balance at the surface (W m$^{-2}$), (c) NH and SH average surface air temperature (°C), (d) NH and SH average sea ice extent ($10^6$ km$^2$), (e) global mean deep ocean temperature averaged below 2500 m (°C), and (f) global mean deep ocean salinity averaged below 2500 m (PSU). The 200

220 years mean value is indicated with a red dash line.

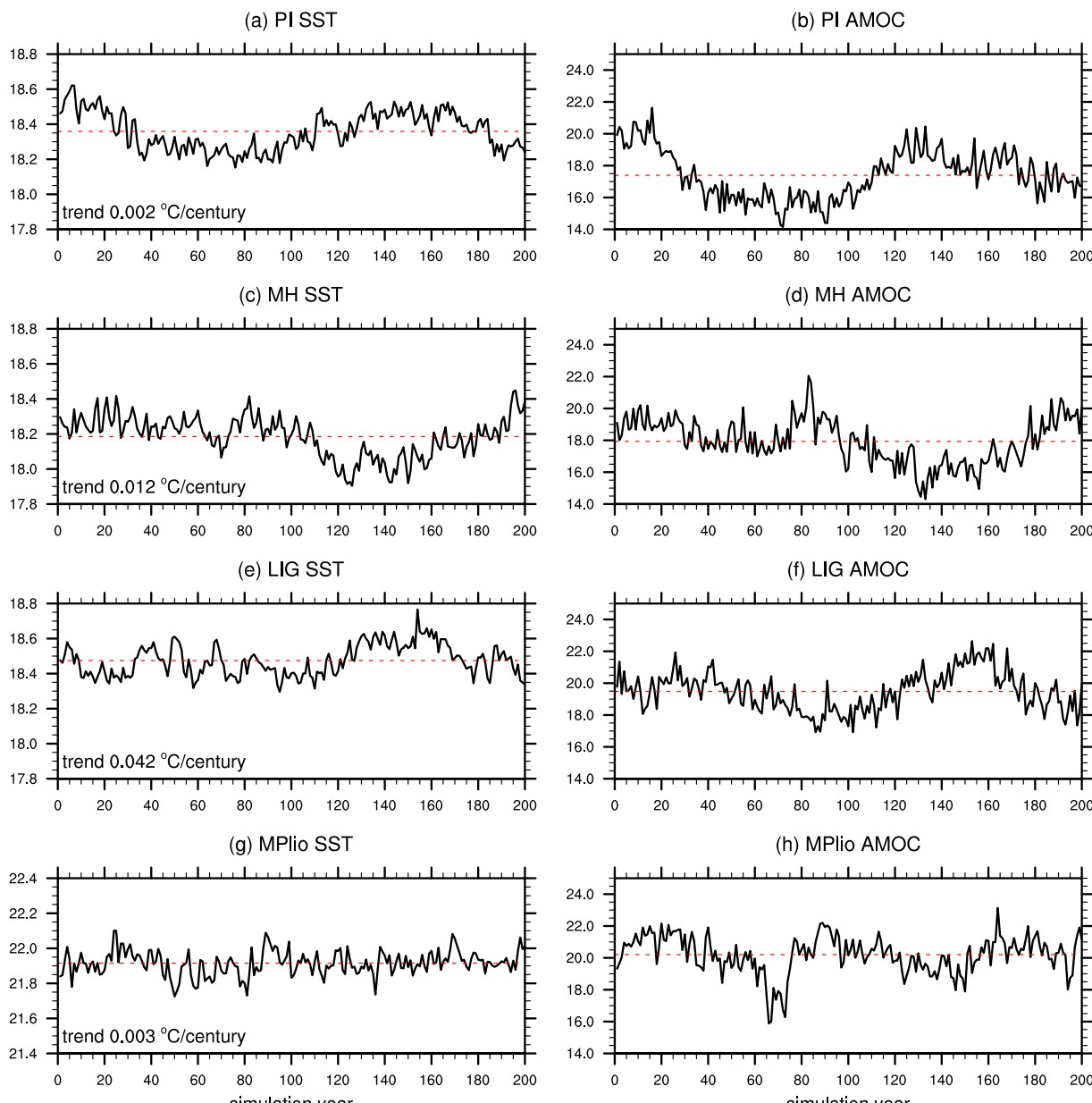

**Figure 4.** Evolution of annual mean globally averaged SST (°C, left panel) and annual mean maximum AMOC (Sv, right panel) in spin-up period for (a, b) the *piControl*, (c, d) the *midHolocene*, (e, f) the *lig127k* and (g, h) the *midPliocene* simulation. The 200 years mean is indicated with a red dash line.

A *piControl* spin-up with EC-Earth3-LR configuration is carried out during the model tunning process and contains minor changes in the simulation process. We apply the initial conditions for the *piControl*, the *midHolocene*, and *lig127k* from the

end of the *piControl* spin-up phase (which ran for approximately 1000 years). Our *piControl* differs from the version of spin-up *piControl* by applying the new implementation of orbital forcing and land-ice physics. Therefore, it takes around another 200 years of spin-up phase for *piControl* simulation to reach equilibrium. The criteria for an equilibrium state are measured by the global mean surface temperature trend < +/- 0.05 °C per century and a stable Atlantic meridional overturning circulation (AMOC) (Kageyama et al., 2018). Due to the changed boundary conditions, the *midHolocene* and *lig127k* simulations reach the equilibrium state after about 200 years of spin-up. The *midPliocene* experiment starts from the initial condition of a Levitus ocean state, and we perform a spin-up for1400 years until it reaches the equilibrium state. The change in the land-sea mask in *midPliocene* requires appropriate initial fields for the changed grid-box. These fields are all modified by the model and are expected to reach equilibrium with the given boundary conditions.

The energy balance at the top of the atmosphere and the surface are stable in all the simulations (a-b in Fig. 3 and Fig S1-S3). We expect a closure of energy in the atmosphere-ocean system for an equilibrium state, and all our simulations show approximately - 0.5 W m$^{-2}$ imbalance (TOA energy minus surface). To reduce this energy imbalance is one primary target in the model tunning process under the pre-industrial climate condition, and the tunned parameters are not allowed to change in any other past or future scenarios according to the CMIP6 protocol. The energy imbalance differs for different resolution tunning. For example, in the EC-Earth3 standard resolution T255L91, the energy imbalance is in the order of 0.25 W m$^{-2}$ (Döscher et al., 2020), and the high-resolution T511L91 tunning version, the imbalance is 0.9 W m$^{-2}$ (personal communication). The 0.5 W m$^{-2}$ energy imbalance in EC-Earth3-LR is in the pre-industrial control simulations range in 25 CMIP5 climate models (Hobbs et al., 2016). One reason may be that the short spin-up as the deep ocean temperature and salinity have shown a larger trend (Fig 3 and Fig. S1-S3) than that of the surface temperature (Fig. 3). The deep ocean temperature has exhibited an increasing trend in all four simulations. The deep ocean salinity shows a decreasing trend in *piControl* (Fig. 3) and *midHolocene* (Fig. S1) but a rising trend in *lig127K* (Fig. S2) and *midPliocene* (Fig. S3). Considering the climates are warmer in the latter two conditions, both sea-ice melting in the polar regions and enhanced oceanic evaporation can influence the surface salinity in different ways. More diagnostics in ocean dynamics are needed to understand these in-depth ocean features in different climate scenarios. No apparent trend is seen in the sea-ice extent in the Arctic and Antarctic (Fig.3 and Fig. S1-S3). The Arctic sea-ice shows a similar low-frequency variability, as seen in AMOC (right panel in Fig. 3), indicating the association between the Arctic sea ice and AMOC. The dramatic overall sea-ice melting is seen in *midPliocene* (Fig. S3) in both the Arctic and Antarctic. In the other three experiments, the Antarctic sea-ice extent remains similar to that of *piControl*. These climate mean features in sea-ice in the spin-up period remains in the production simulation, which is performed from the end of each spin-up. We present the large-scale features from 200 years of production simulations in section 4.

We present the time evolution of the global mean sea surface temperature (SST) and the maximum AMOC in Fig. 4 for the four simulations. To be comparable, we show 200 years spin-up period for all the four experiments. The trends in global mean sea surface temperature in the four spin-up have met the criteria of equilibrium. In the EC-Earth3-LR simulations, the

maximum AMOC mostly appears at 25° N latitude, and occasionally shifts northward to 45° N. AMOC is about 17.5 Sv *piContro* simulation and becomes stronger in the *midHolocene, lig127k*, and *midPliocene* simulations. The multidecadal variability is apparent in both AMOC and SST in the *piControl,* the *midHolocene* and *lig127k* simulations. The *midPliocene* is the warmest period and show less low-frequency variability.

## 3.5 Post-processing of the model output for PMIP4-CMIP6

Following the CMOR tables (Climate Model Output Rewriter, output format under all CMIP standards), we use the ece2cmor3 software package for post-processing model output. Ece2cmor3 is a python package developed by the EC-Earth community to convert the EC-Earth3 model output to a CMIP6 compliant format. It uses climate data operators (CDO) (Schulzweida, 2019) and the CMOR library bindings to select variables and vertical levels, perform time-average (or take extreme daily), interpolate spectral and grid-point atmospheric fields to a regular Gaussian grid, and calculate variables by an arithmetic combination of the original model fields. The ece2cmor3 package uses the PCMDI CMOR-library to produce netCDF files with the appropriate format and metadata. All the data presented in this paper are publicly available at CMIP6 database at https://esgf-node.llnl.gov/projects/cmip6/.

It is worth noting that particular caution should be paid for downloading the *piControl* data from ESGF data nodes. There are several *piControl* by the EC-Earth3 model available in the CMIP6 database. It is appropriate to select the one with the model name EC-Earth3-LR as the other PMIP4 experiments' reference. The other *PiControl* experiments by EC-Earth3 differ in the model configuration or model resolution.

## 4. Climate responses in the *midHolocene*, *lig127k* and *midPliocene* simulations

Here we present the changes in the large-scale features of the three past warm climates simulated by the EC-Earth3-LR, i.e., the two interglacial simulations *midHolocene* and *lig127k*, as well as the high $CO_2$ featured *midPliocene,* with the comparison to the reference simulation *piControl*.

## 4.1 Large scale climate response

We summarise the global annual mean values for climate anomalies from the three experiments to the *piControl* in Table 3. The global mean temperature from the *midHolocene* shows a slightly cooling, placing the simulation towards the upper end of the PMIP4 *midHolocene* ensemble (Brierley et al., 2020). The global mean precipitation anomaly is mild in the *midHolocene*, with about 5-7 % more precipitation than in *piControl* over the land during the boreal summer and autumn, following the enhanced Northern Hemisphere monsoon (Sect. 4.5). The more considerable changes in orbital forcing in the *lig127k* results in 0.4 °C warmings globally, and this warming is amplified over the land and in the Arctic. In the *lig127k,* the land precipitation

increased 16 % in boreal summer, indicating a more vigorous monsoon than in the *midHolocene*. The *midPliocene* shows significant global warming reaching 4.8 °C, which is among the warmest simulations with the highest Arctic amplification in the PlioMIP2 ensemble (Haywood et al., 2020b;de Nooijer et al., 2020b). The precipitation increases by more than 10 %
globally, and the land precipitation increased more than 20 % in boreal summer and autumn season in the *midPliocene*.

**Table 3.** Global mean temperature and precipitation anomalies with respect to the *piControl*, for annual mean (ANN), December-January-February mean (DJF), March-April-May mean (MAM), June-July-August mean (JJA), and September-October-November (SON).

| Experiment | Season | Surface air temperature anomalies | | | | | Precipitation anomalies | | |
| --- | --- | --- | --- | --- | --- | --- | --- | --- | --- |
| | | | | | | | mm/day (per cent change) | | |
| | | Global | Land | Sea | 60°N-90°N (75°N-90°N) | 60°S-90°S (75°S-90°S) | Global | Land | Sea |
| *midHolocene* | ANN | -0.1 | 0 | -0.1 | 1.3 (1.4) | 0.1 (-0.1) | 0 (0) | 0.04 (1.7) | -0.02 (-0.6) |
| | DJF | -0.2 | -0.4 | -0.1 | 1.1 (1.1) | -0.08 (-0.54) | 0.03 (1) | -0.03 (-1.2) | 0.07 (2.2) |
| | MAM | -0.7 | -0.9 | -0.5 | 0.2 (0.1) | -0.42 (-0.82) | -0.02 (-0.7) | -0.07 (-3.1) | 0 (0) |
| | JJA | 0.2 | 0.5 | 0 | 1.6 (1.9) | -0.02 (0) | -0.02 (-0.7) | 0.12 (5.0) | -0.10 (-3.2) |
| | SON | 0.5 | 0.7 | 0.4 | 2.1 (2.7) | 0.72 (1.04) | 0.02 (0.7) | 0.15 (6.6) | -0.06 (-2.0) |
| *lig127k* | ANN | 0.4 | 0.7 | 0.2 | 3.6 (4.3) | -0.14 (-0.15) | 0.03 (1.0) | 0.10 (4.3) | -0.01 (0.3) |
| | DJF | -0.6 | -0.8 | -0.5 | 2.8 (3.4) | -0.99 (-1.70) | 0.03 (1.0) | -0.17 (-7.2) | 0.14 (4.4) |
| | MAM | -0.3 | 0.1 | -0.5 | 2.9 (3.1) | -0.82 (-1.12) | -0.06 (-2.1) | 0.06 (2.6) | -0.13 (4.2) |
| | JJA | 1.7 | 3.0 | 1.0 | 5.0 (4.8) | 0.03 (0.15) | 0.03 (1.0) | 0.38 (15.9) | -0.17 (5.4) |
| | SON | 0.8 | 0.6 | 0.9 | 3.6 (5.8) | 1.23 (2.10) | 0.12 (4.3) | 0.12 (5.2) | 0.12 (3.9) |
| *midPliocene* | ANN | 4.8 | 6.3 | 4.0 | 11.6 (13.1) | 8.28 (10.4) | 0.31 (11.0) | 0.45 (19.4) | 0.23 (7.4) |
| | DJF | 4.6 | 5.9 | 3.9 | 10.6 (12.6) | 6.91 (8.33) | 0.27 (9.4) | 0.37 (15.7) | 0.21 (6.7) |
| | MAM | 4.4 | 6.1 | 3.8 | 10.8 (10.3) | 8.45 (11.61) | 0.29 (10.4) | 0.41 (18.1) | 0.21 (6.7) |
| | JJA | 4.9 | 6.5 | 4.0 | 11.0 (9.5) | 9.71 (11.85) | 0.35 (12.3) | 0.49 (20.6) | 0.26 (8.3) |
| | SON | 5.13 | 6.70 | 4.2 | 14.1 (20.0) | 8.05 (9.86) | 0.33 (11.9) | 0.51 (22.4) | 0.23 (7.5) |

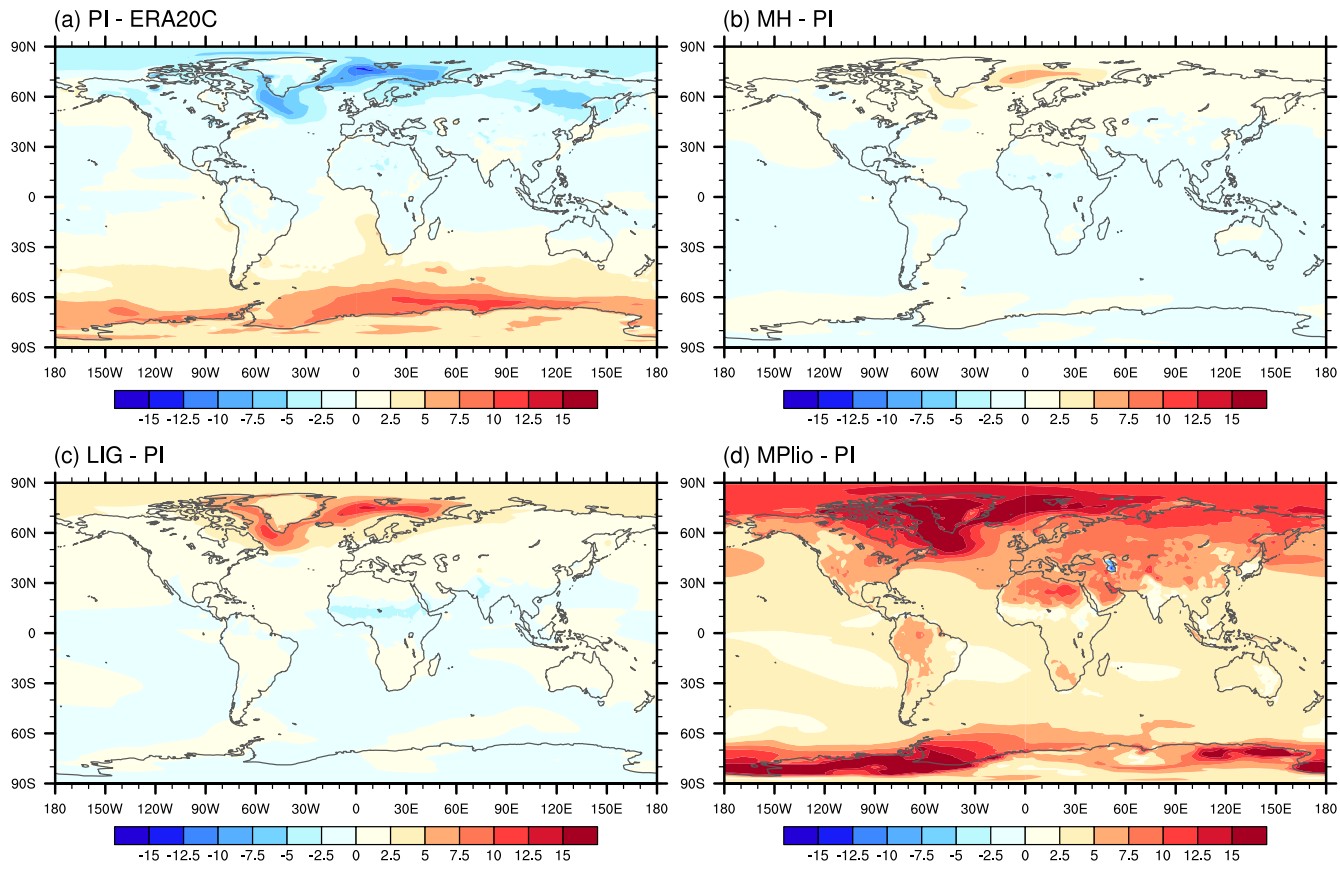

**Figure 5**. Mean annual temperature (°C) in (a) difference between the *piControl* and ERA20C (1901-1930 mean), and the anomaly in the (b) the *midHolocene*, (c) the *lig127k* and (d) the *midPliocene* from the *piControl*.

For the climate changes in global pattern, we show a comparison between the annual mean temperature and precipitation in *piControl* with the early century (1901-1930) mean in ERA20C reanalysis data (Fig. 5a and Fig. 6a). The EC-Earth3-LR tends to overestimate the temperature in high latitudes, with cold biases in NH and warm biases in SH. This is a reasonably robust feature of most global climate models, particularly prominent in the EC-Earth model and the cause attributed to the simulated sea ice (Koenigk et al., 2013;Chapman and Walsh, 2007). The dry biases in NH and wet biases in SH are noticed when comparing the simulated PI precipitation with ERA20C, primarily in the equatorial ocean area (Fig. 6a). In the following analysis, we compare the anomalies in the three paleoclimate experiments to the *piControl*; any changes in the temperature reflect the responses to the external forcing.

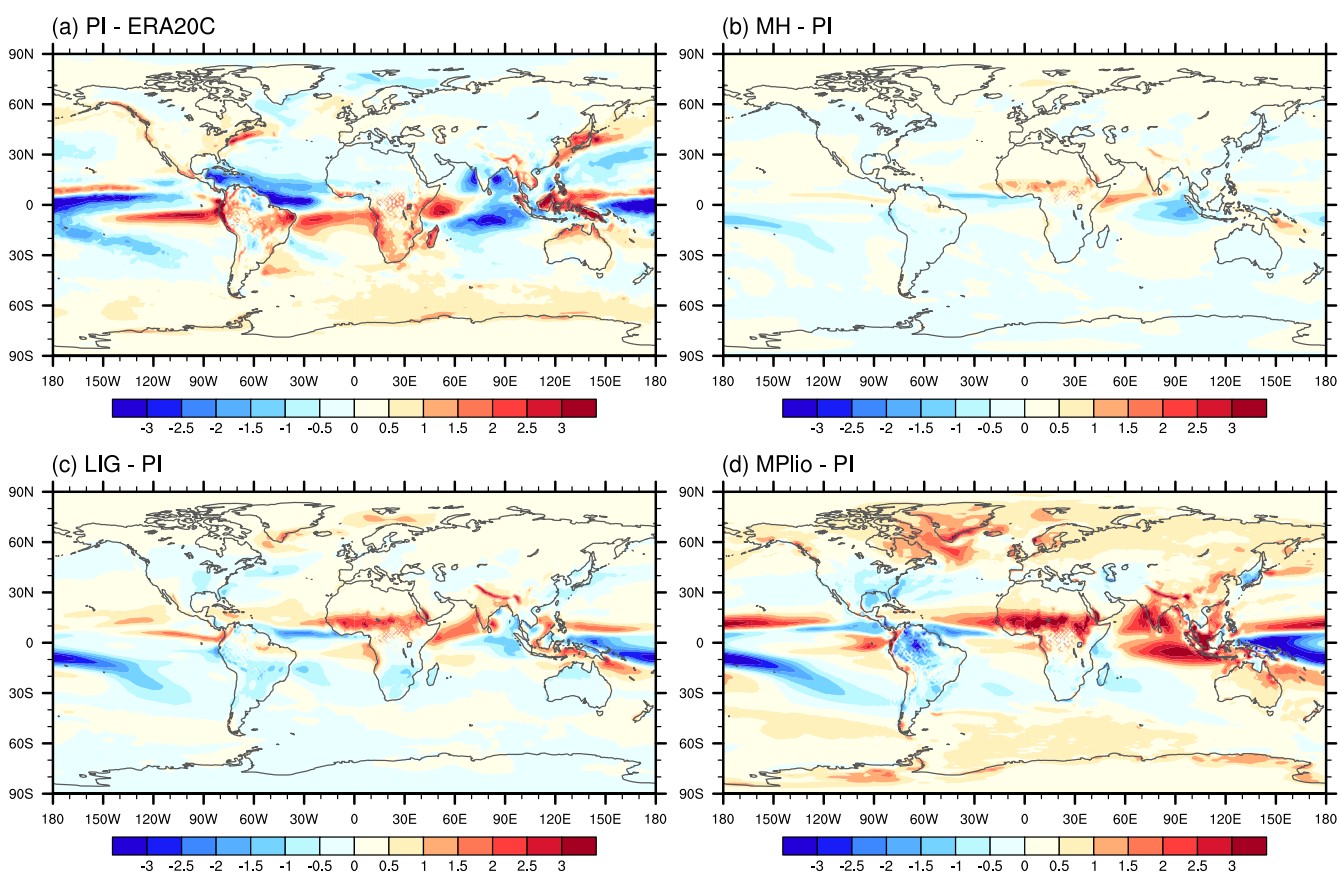

**Figure 6**. Mean annual precipitation (mm d$^{-1}$) in (a) difference between the *piControl* and ERA20C (1901-1930 mean), and the anomaly in the (b) the *midHolocene*, (c) the *lig127k* and (d) the *midPliocene* from the *piControl*.

During the two interglacial periods, the orbital forcing is characterized by the redistribution of the insolation over the globe in different seasons (Fig. 1). This does not affect too much of the global energy balance; therefore, global mean climate changes are insignificant (Table 3). The spatial patterns of mean annual temperature show warming in the Arctic region in the two interglacial periods relative to the pre-industrial, with a larger magnitude in the *lig127k* than in the *midHolocene* (Fig. 5b and

Fig. 5c). The warming in southern high-latitudes shown in the PMIP4 *midHolocene* ensemble (Brierley et al., 2020a) is not apparent in EC-Earth3-LR simulation. The simulated *midPliocene* warming is much stronger than those of the two interglacial periods, dominated by the robust Arctic amplification in EC-Earth3-LR (de Nooijer et al., 2020b). In the *midPliocene,* the strong warming amplification appears in both polar regions (Fig. 5d), which contrasts the weak warming in some parts in the southern high-latitudes in the *midHolocene* and *lig127K* (Fig. 5b and Fig. 5c). The seasonal features of the temperature

response in the polar regions are discussed more in detail in Sect. 4.4. In all the three warm periods, the response in annual

mean temperature is stronger over the land than over the ocean (Fig. 5 and Table 3), which is a common feature in most PMIP4-CMIP6 simulations (Brierley et al., 2020a;Otto-Bliesner et al., 2020;Haywood et al., 2020b). These notable changes in polar amplification and land-sea contrast modulate the temperature gradients and lead to changes in the large-scale circulation (Sect. 4.4) and global monsoons (Sect. 4.5). The most pronounced and robust changes in the global land precipitation occur in the Sahel and the subcontinent of India, indicating enhanced western African monsoon and Indian monsoon in the *midHolocene*, *lig127k* and *midPliocene* (Fig. 6). The increased East Asian monsoon appears in *midPliocene* but not evident in *midHolocene* and *lig127k*. More analyses on global monsoon are presented in Sect. 4.6. Over the Indian ocean, the precipitation increases over the western Indian Ocean. It decreases in the east Indian Ocean during *midHolocene* (Fig. 6b) and *lig127k* (Fig. 6c)., opposite to those in *midPliocene* (Fig. 6d). There is a northward shift of the ITCZ in the Atlantic and Pacific rain belts in the three simulations, with magnitude increases in *midHolocene*, *lig127k* and *midPliocene*. Increased precipitation is also observed in northern high-latitudes in the three warm periods associated with the strong Arctic warming amplification.

## 4.2 Data-model comparison

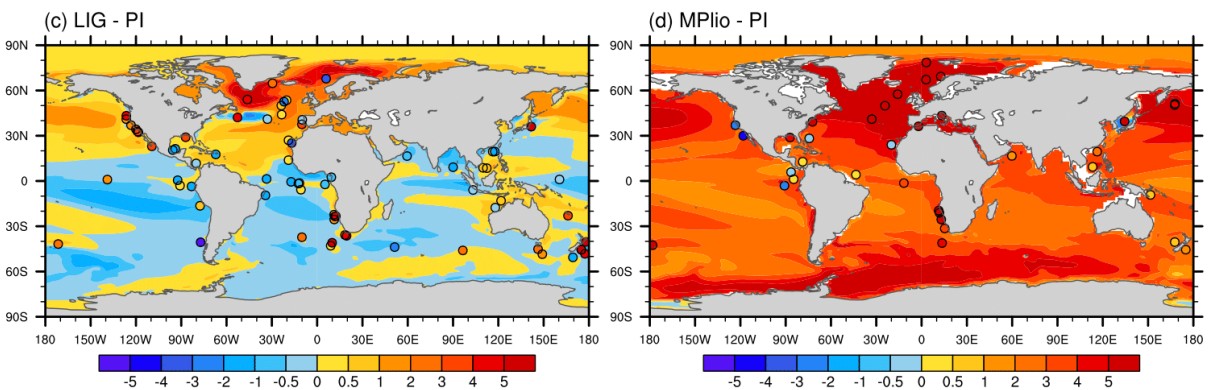

Figure 7.   Mean annual sea surface temperature (°C) anomalies between a) lig127k and b) *midPliocene*, and the *piControl*. The shadings are overlaid by the proxy-data anomalies from a) Hoffman et al. (2017) and b) Foley and Dowsett (2019), calculated relative to HadISST (1870-1899 mean). Proxy data anomalies for the data points are described in Section 4.2 for lig127k and midPliocene, respectively.

The model simulations for *lig127k* and *midPliocene* are assessed using two proxy-reconstructions of sea surface temperatures, following the methods presented in Williams et al. (2020) and Haywood et al. (2020a). Due to the low spatial coverage of available SST-reconstructions for the *midHolocene*, this period was excluded from the data-model comparison. The *lig127k* simulation is assessed against the global marine compilation by Hoffman et al. (2017), consisting of 86 annually reconstructed multi-proxy records from forams, radiolaria, coccolithophore, alkenones, Mg/Ca and diatoms, and the values are taken from

the provided 127 ka time-slice. The *midPliocene* simulation is assessed against the SST data set provided by Foley and Dowsett (2019) covering a 30,000 years interval around the MIS KM5c peak. The proxy records are reconstructed using the alkenone paleotemperature (Uk37) method. The anomalies for both *lig127k* and *midPliocene* are calculated relative to pre-industrial, 1870-1899 HadISST mean SSTs (Rayner et al., 2003) for the same data-points.

In Fig. 7, we show the mean annual sea surface temperatures anomalies from the two model simulations, calculated with respect to the *piControl*. The *lig127k* (Fig. 7a) show a warming of the Northern Hemisphere with the highest values in the Arctic region. In contrast, the Southern Hemisphere shows a widespread cooling, overall consistent with the PMIP4 ensemble (Otto-Bliesner et al., 2020). There is generally good agreement globally between the *lig127k* simulation and the reconstructed SSTs. The proxies indicate high-latitude warming in both hemispheres and cooling along the Equator, which is consistent with the overall pattern of the model simulation. However, the magnitude of warming shown by the proxies in the Southern Hemisphere suggests larger warming than is captured in EC-Earth3-LR. Over the North Atlantic, proxy-reconstructions indicate warming right next to cooling, making a comparison to the simulation difficult, but this suggests that EC-Earth3-LR overestimates the homogeneity of the North Atlantic warming in the *lig127k* and fails to capture both the cooling and its magnitude in the Norwegian Sea. However, this feature is consistent with the results of the PMIP4 ensemble, which also fails to capture this cooling (Otto-Bliesner et al., 2020). The magnitude of warming of the coasts in the Southern Hemisphere (e.g., west of South Africa and east of New Zealand) shows large discrepancies between the model simulation and proxy reconstructions in signs of change in magnitude especially. Simulated *lig127k* SST suggests temperature anomalies of up to 0.5°C while the proxy dataset indicates changes exceeding 5°C. This underestimation of the warming of coastal upwelling regions is seen in the PMIP4 ensemble. It is suggested to result from a warm bias in the *piControl*-simulations leaving little room for warming and the low resolution in the PMIP4 ensemble models failing to capture these narrow regions (Otto-Bliesner et al., 2020).

The *midPliocene* simulation (Fig. 7b), shows general warming globally. A strong polar amplification appears in both hemispheres, consistent with the PlioMIP2 ensemble (Haywood et al., 2020a;de Nooijer et al., 2020a) and the proxy dataset (Foley and Dowsett, 2019). The strong North Atlantic warming seen in proxies are captured well by the model simulation, both spatially and in its magnitude. However, EC-Earth3-LR overestimates the warming around the equatorial belt and to some degree also the sign of change, especially in the Central American region where proxy records suggest some degree of cooling during *midPliocene*. Most considerable discrepancies are found off the coast of California, with up to 8°C difference between model and proxy data, and in the Sea of Japan where simulated SST shows a cooling while the proxy data suggests warming.

### 4.3 Response in Hadley circulation and Walker circulation

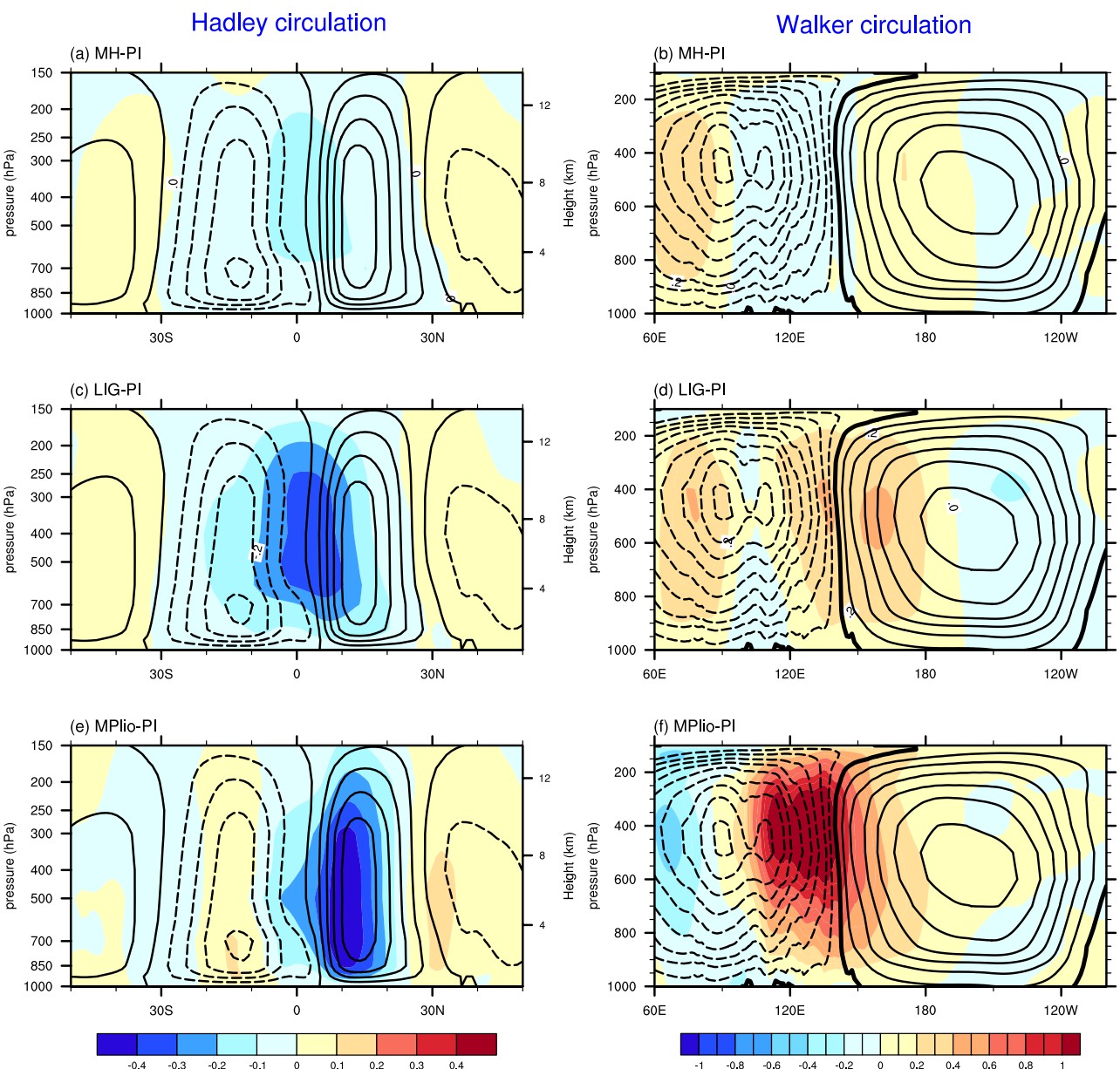

**Figure 8**. The changes in annual mean Hadley circulation (represented by zonal-mean meridional mass stream function in the left panel) and Walker circulation (represented by zonal mass stream function averaged between 10º S~10º N over Indo-Pacific region 60º E-150º W in the right panel). The shadings indicate the changes in (a, b) the *midHolocene*, (c, d) the *lig127k* and (e, f) the *midPliocene* respect to *piControl*. The contours represent the climatology from the *piControl* simulation (solid contours indicate positive value and dashed contours indicate negative value). Units: $10^{10}$ kg s$^{-1}$.

The changes in global temperature pattern, shown above, could induce large atmospheric circulation anomalies, i.e., in the Hadley circulation and Walker circulation. Previous studies have demonstrated that the meridional surface air temperature gradient between low latitude and mid-to-high latitude determine the changes in Hadley circulation (Corvec and Fletcher, 2017;Burls and Fedorov, 2017). Indeed, the weakening of the Hadley circulation seen in the NH in the three experiments (Fig. 8a, 8c and 8e) corresponds to the decreased temperature gradient between tropics and mid-to-high latitude (Fig. 8b-d). However, the thermal contrast is less pronounced in the Southern Hemisphere in the *midHolocene* and the *lig127k* simulations. The two simulations even show slight cooling in the SH compared to the tropics (Fig. 4b and 4c), resulting in a slightly stronger meridional temperature gradient, and induce stronger Hadley circulation (Fig. 8c). The similar results can be seen in Fig. 9a, all of the three simulations show a weaker Hadley circulation in both Hemisphere, except the *lig127k* and *midHolocene* in southern Hemisphere. All three warm period simulations show the Hadley circulation is roughly poleward shift around 30°N/20°S, with respect to PI simulation. In the *lig127k* and *midPliocene* simulations, we observe a poleward widening of Hadley circulation in both Hemisphere. This widening is less evident in the *midHolocene* simulation.

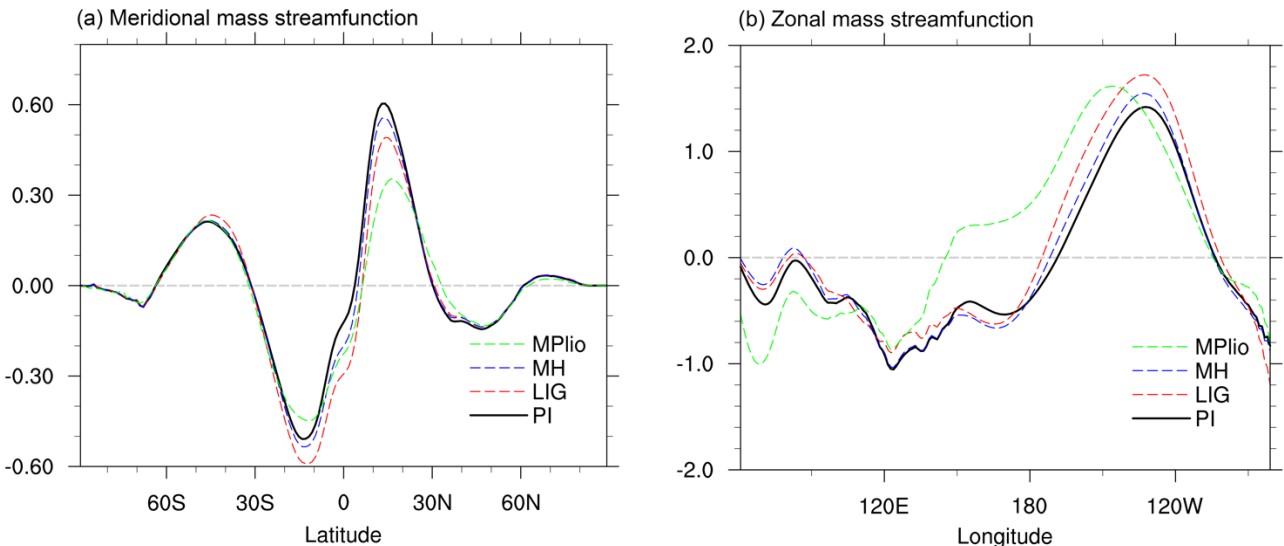

**Figure 9**. The meridional (a) and zonal (b) mean of vertical integrated meridional and zonal mass stream function in Figure 5. Unit: $10^{10}$ kg s$^{-1}$.

The Walker circulation in *piControl* is characterized as ascending in the maritime continent and western Pacific and descending in the eastern Pacific (contours in Fig. 8b, 8d and 8e), which is consistent with previous studies (Peixoto and Oort, 1992;Kamae et al., 2011;Bayr et al., 2014). Compared to the *piControl*, all three warm period simulations show strong ascending over the tropical western Pacific, indicating a strengthening and westward shift of the Pacific Walker circulation cell (Fig. 8b, 8d and

8f). These features are further shown in Fig. 9b, and it is clear that all three simulations indicate a stronger and westward shift of Pacific Walker circulation. There is more westward shift in the *lig127k* and *midPliocene* than the *midHolocene* simulation, corresponding to less ascending over the maritime continent. The increasing magnitude of shift from *piControl* to *midHolocene*, *lig127k* and *midPliocene* can be associated with reduced ENSO variability (Sec 4.6).

## 4.4 Climate responses in the polar region

The poles' response to the different boundary conditions is of particular interest as the largest temperature anomalies are observed here (Fig. 4), which are known to associate with strong feedback at the poles such as ice-albedo and carbon cycle feedbacks (Masson-Delmotte et al., 2013). The amplification of global temperature anomalies in the Arctic is a near-universal feature of historical simulations in response to the increased greenhouse gas. It is also observed in the instrumental record (Serreze and Barry, 2011). Similar amplification of global temperature anomalies is not observed in the Antarctic, which may be attributed to the rapid removal of surface heat in the Southern Ocean, which, in turn, limits the ability of climate feedbacks to amplify the warming (Stroeve et al., 2007). Arctic amplification is prominent in the EC-Earth3-LR *midPliocene* simulation, with Arctic (60-90° N) temperature anomalies exceeding global temperature anomalies by a factor 2.4 (Table 3). The Arctic warming in the *midHolocene* and *lig127k* simulations are not as striking as in the *midPliocene*. Given that the global mean temperature anomalies in the *midHolocene* and *lig127k* simulations are small (Table 3), it is inappropriate to apply an amplification ratio as discussed in Hind et al. (2016).

The minimum Arctic surface air temperature (SAT) warming is expected in the summer because of increased ocean heat uptake following reductions in sea ice, while maximum SAT warming is expected in the (boreal) autumn and winter following the release of this heat (Pithan and Mauritsen, 2014;Serreze et al., 2009;Yoshimori and Suzuki, 2019;Zheng et al., 2019). This expected seasonality of SAT is not present in the Arctic in the *midHolocene* and *lig127k* simulations (Fig. 10). There are peaks in Arctic SAT warming, amounting to +2.3° C and +5.8° C in the *midHolocene* and *lig127k* simulations, respectively, in the boreal summer (Fig. 10a) due to positive insolation changes (Fig. 1). Negative insolation changes during the autumn may offset some of the expected autumn excess heat release. In the *midPliocene* simulation, Arctic SAT warming peaks in the autumn at +15.2° C, but there is no exact summer minimum (Fig. 10a). In the Antarctic, the SAT seasonality in the two interglacial simulations does not differ significantly from the *piControl* simulation. In the *midPliocene*, Antarctic warming peaks in late austral winter and early spring (July-September) with maximum warming reaching +10.4° C in August.

Arctic SAT warming will lead to a reduction in the sea ice extent. The sea ice extent defines a region as either "ice-covered" or "not ice-covered." In the model grid, for each grid cell, either the cell has ice (usually a value of "1") or the cell has no ice (usually a value of "0"). Here we apply a commonly used threshold 15% (such as used by National Snow and Ice Data Center NSIDC) to determine the ice labelling, meaning that if the model grid cell has greater than 15% ice concentration, the cell is

labelled as "ice-covered." Compared with the *piControl*, the simulations show significant decreases in sea ice extent (SIE) with anomalies in the Arctic peaking at -1.8 x $10^6$ km$^2$ in the *midHolocene*, -5.2 x $10^6$ km$^2$ in the *lig127k*, and -11.8 x $10^6$ km$^2$ in the *midPliocene*. SIE anomalies peak earliest in the year in the *lig127k* simulation, followed by the *midHolocene* (Fig. 10c). In the *midPliocene* simulation, the Arctic experiences sea ice-free conditions from August to October, and anomalies peak both in the (boreal) autumn and the spring. The different timing of the SIE anomaly peaks may be explained by a runaway sea

ice-albedo feedback (Feng et al., 2019) in which early summer reductions in SIE lead to more warming through decreased albedo and, in turn, even more significant reductions in SIE. Since the positive insolation changes in the Arctic occur earlier in the year in the *lig127k* than in the *midHolocene*, this feedback would be triggered earlier in the year in the *lig127k*. The importance of this feedback depends on the annual amplitude of SIE. Since this is relatively small in the *midPliocene*, it results in the weakest link between the seasonal SIE and SAT cycles. This may explain the absence of a distinct minimum in Arctic

SAT anomalies in the boreal summer in the *midPliocene* simulation. The seasonal cycle of Antarctic SIE anomalies closely resembles the SAT cycle, and anomalies peak in September at -9.2 x $10^6$ km$^2$ in the *midPliocene* simulation. Anomalies are around zero throughout the year in the *midHolocene and lig127*, consistent with the limited changes in SAT observed at these latitudes throughout the year.

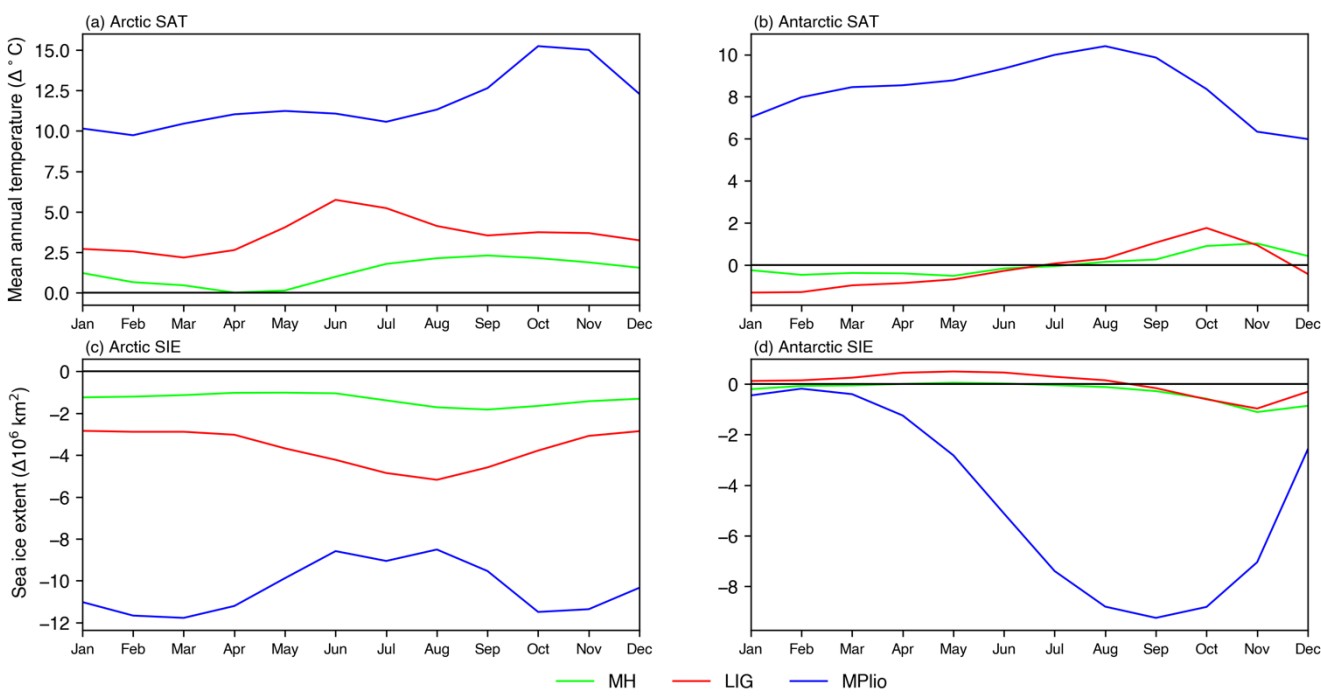

**Figure 10**. Seasonal changes of Arctic mean surface air temperature (a and b) and sea ice extent (c and d) in the Arctic (a and c, 60° N-90° N average), and Antarctic (b and d, 60° S-90° S average) for each simulation.

The sea ice edge in the months of maximum and minimum SIE in each simulation is depicted by Fig. 11. The maximum SIE in the Arctic is observed in March for each simulation (Fig. 11a). For the Arctic minimum SIE, observed in August, the sea ice edge is limited to the Greenland Sea for the *piControl* and *midHolocene* simulations in the Atlantic domain, while it does not reach farther than the Fram Strait in the *lig127k* simulation (Fig. 11c). In the Antarctic, the maximum SIE is in September (Fig. 11b). Sea ice is present along most of Antarctica's coast during September in the *midPliocene* simulation (Fig. 11b). The minimum SIE is observed in February. The sea ice edge is mainly limited to the Weddell Sea for the *piControl*, *midHolocene*, and *lig127k* simulations, while no sea ice is present in the *midPliocene* simulation (Fig. 11d). The *midPliocene* SIE anomalies are the lowest in the summer because low sea ice concentrations during this season limit SIE decrease in the mid-Pliocene, while SIE can still decrease more in the PI simulation.

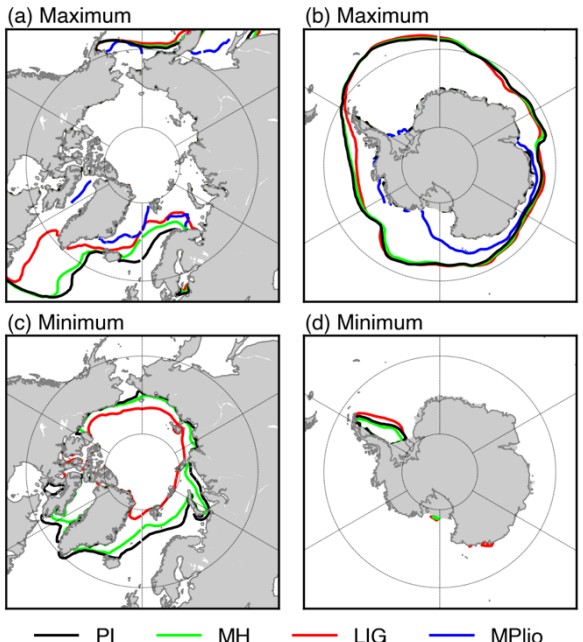

**Figure 11.** Sea ice edge in the Arctic (a, c) and Antarctic (b, d) in the maximum (a, b) and minimum (c, d) SIE month in each simulation. Maximum sea ice extents occur in March in the Arctic, and in September in the Antarctic, while minimum sea ice extents are seen in August in the Arctic, and in February in the Antarctic.

## 4.5  Responses in global monsoon systems

We compare the global monsoon area in different simulations in Fig. 12. Here the global monsoon area is defined as the regions where the annual range precipitation exceeds 2 mm day$^{-1}$, and the local summer precipitation exceeds 55% of the annual precipitation (Liu et al., 2009). The annual range refers to the precipitation difference between MJJAS (May to September)

and NDJFM (November to March) in the NH, and difference between NDJFM and MJJAS in the SH. The precipitation averaged for the five months over the monsoon area represents the global monsoon intensity (Zhou et al., 2008).

There is little difference in land monsoon regions between the *piControl* and *midHolocene* (Fig. 12b), but still an increased area globally by 9 % in the *midHolocene* with the majority of this increase (85 %) located in the SH (Fig. 12b and Fig. 12a). The most significant difference is found in the Southern Indian Ocean, where South African and Australian monsoon regions meet (Fig. S4). However, the SH monsoon intensity decreases in the same period (Fig. 13b). The expansion in this monsoon region is not linked to any visible change in summer temperature or sea level pressure (Fig. S5a and Fig. S5b). It is likely

caused by the precipitation decrease over the eastern Indian Ocean during the SH winter being more extensive than the reduction during the SH summer, leading to stronger seasonal differences (Fig. S4a and Fig. S4b).

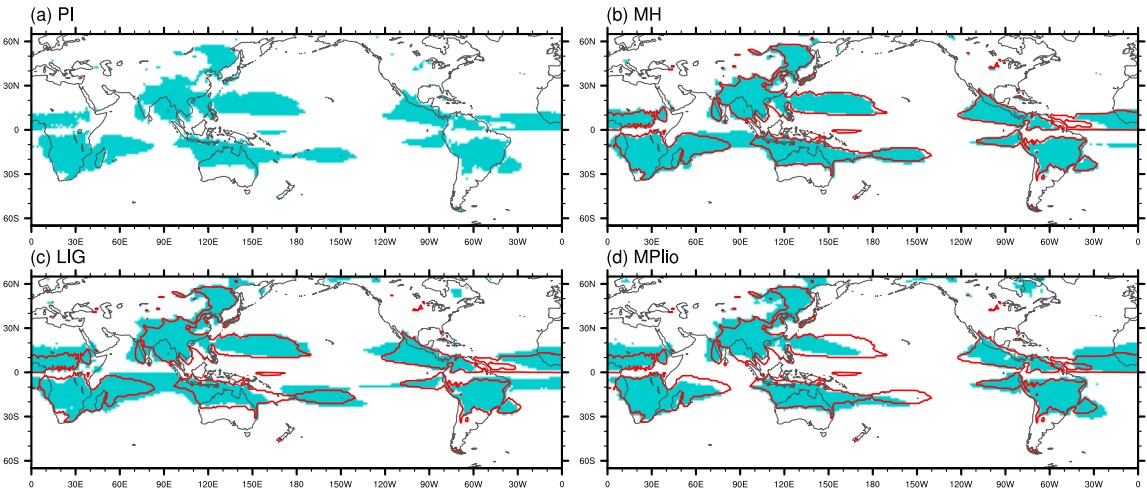

**Figure 12**. Global monsoon area in (a) the *piControl*, (b) the *midHolocene*, (c) the *lig127k*, (d) the *midPliocene* simulation, with *piControl* global monsoon boundary as a red line in b-d.

The Last Interglacial experiences northward expansions of NH monsoon regions (northern Africa, Asia and North America) compared to pre-industrial. In contrast, the expansions in the SH monsoon regions are mainly located over the ocean (Fig.

12c). Although the monsoon area increases in both hemispheres, the intensity has only increased in the NH and decreased in the SH (Fig. 13b). MJJAS exhibits positive temperature anomalies and negative pressure anomalies over land in *lig127k*, favour a strengthening of monsoons (Webster et al., 1998). NDJFM experiences a global decrease in temperature except for the Arctic region, positive pressure anomalies over land and negative pressure anomalies over the ocean, leading to a weakening of the negative land-ocean pressure gradients often linked to monsoonal flows. This result is consistent with the

PMIP4-CMIP6 *lig127k* ensemble, showing that the insolation changes during the *lig127k* lead to an intensification of boreal summer monsoons in the NH and a weakening of the austral summer monsoons in the SH (Otto-Bliesner et al., 2020).

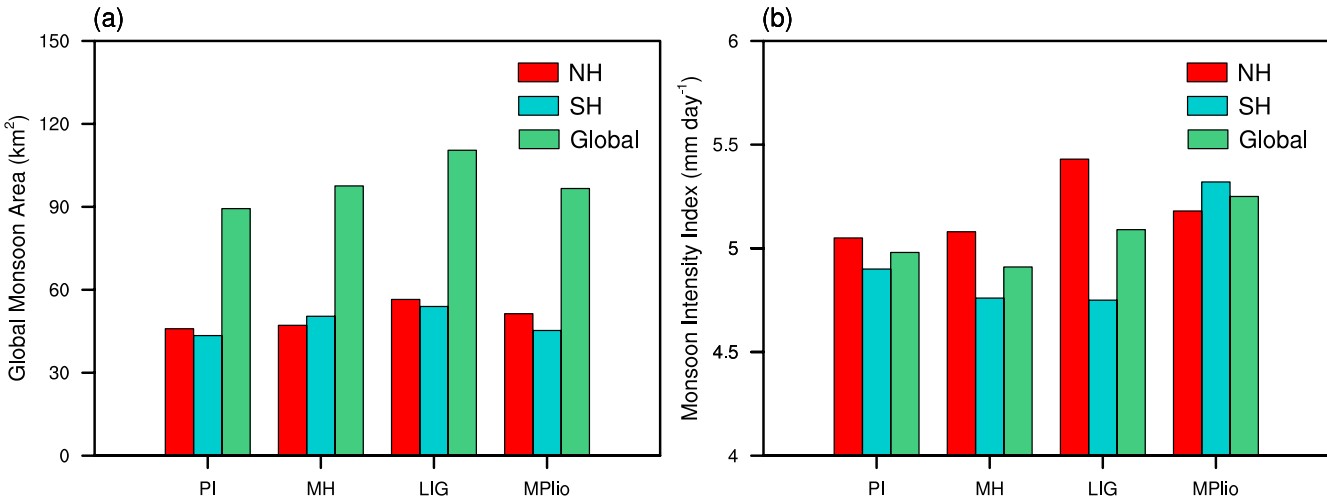

**Figure 13**. Global monsoon area (a) and Monsoon intensity (b) for global, the NH and the SH in the four simulations.


The global monsoon area increases with of 8 % during the *midPliocene*, which comes mainly from the NH (75 %), while the increase in monsoon intensity comes mostly from the SH (9 % in the SH compared to 3 % in the NH) (Fig. 13). Compared to the other two interglacial periods, *midPliocene* is the only period with a higher monsoon intensity in the SH than in the NH. All NH monsoon regions except North America experience a northward expansion in MJJAS (Fig. 12d), consistent with
strengthened latitudinal and land-sea temperature contrast and deepening of the low-pressure areas over Eurasia, North Africa and the North Atlantic (Fig. S4e). It is also consistent with the northward expansion of Hadley circulation (Fig. 8e). The precipitation intensity increase found in SH monsoon regions (NDJFM) is not linked to any strengthened land-sea pressure gradients (Fig. S4f). It might result from both a slightly strengthened land-sea thermal gradient for South America and southern Africa and a general temperature increase leading to higher specific humidity and increased moisture transport to the monsoon
regions.

**4.6    Responses in the modes of climate variability**

Lastly, we examine the simulated climate variability modes on interannual and longer timescales associated with changes in sea surface temperature patterns, namely, the El Niño-Southern Oscillation (ENSO), the Pacific Decadal Oscillation (PDO) and the Atlantic Multidecadal Oscillation (AMO). We define the interannual variability of monthly SST anomalies in the
tropical Pacific basin as ENSO spatial pattern, and the leading EOF1 time series as ENSO index. The PDO is calculated by

the leading EOF of monthly SST anomalies in the North Pacific and PC1 time series following (Mantua and Hare, 2002). Similarly, the AMO is estimated by the leading EOF of monthly SST anomalies in the North Atlantic (Guan and Nigam, 2009).

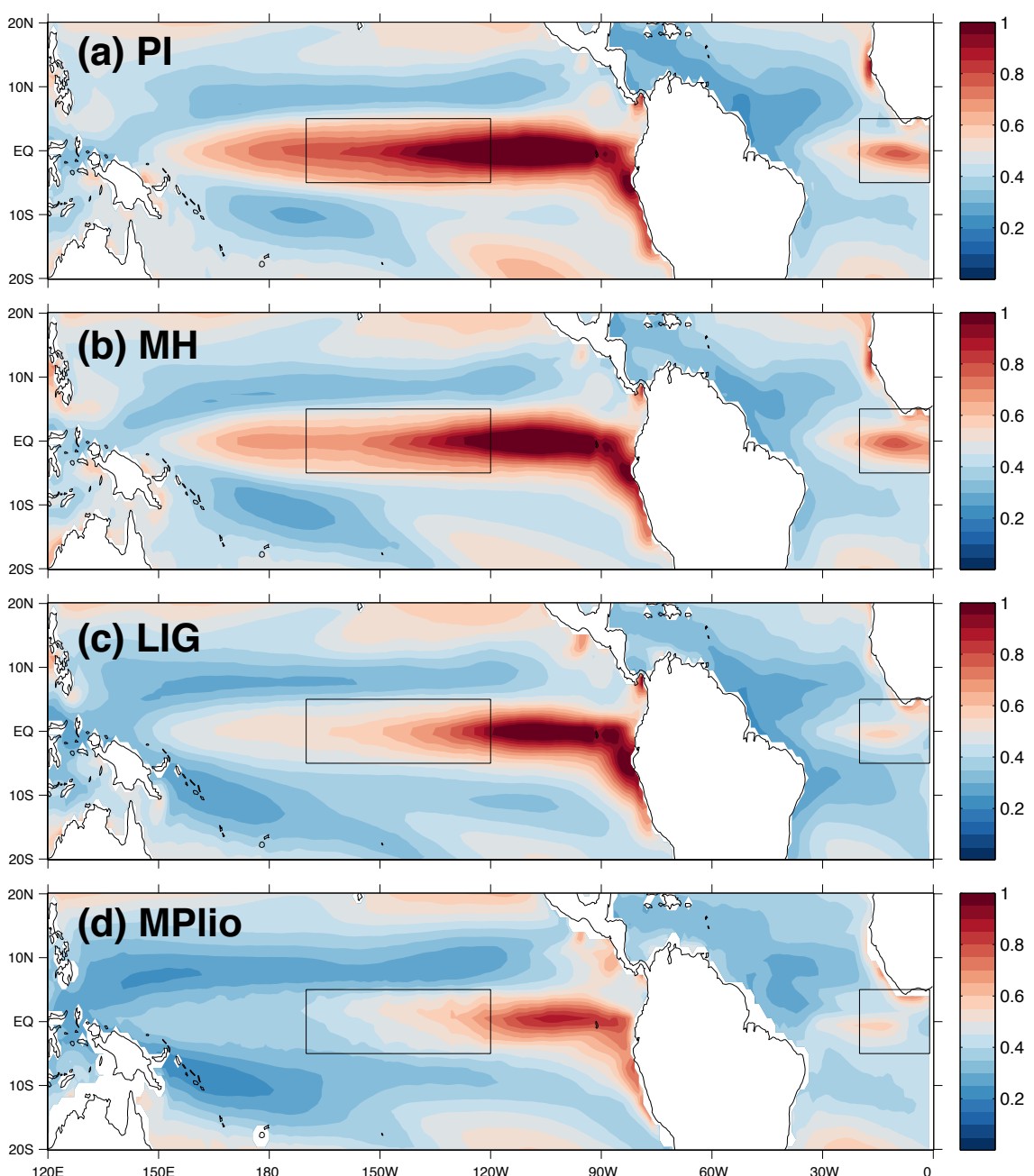

**Figure 14**. Simulated SST interannual variability (standard deviation of SST anomalies after removing seasonal cycle, unit: °C) for all simulations. Black boxes are Niño 3.4 and Atlantic 3 regions (for the El Niño-Southern Oscillation and Atlantic Niño, respectively).

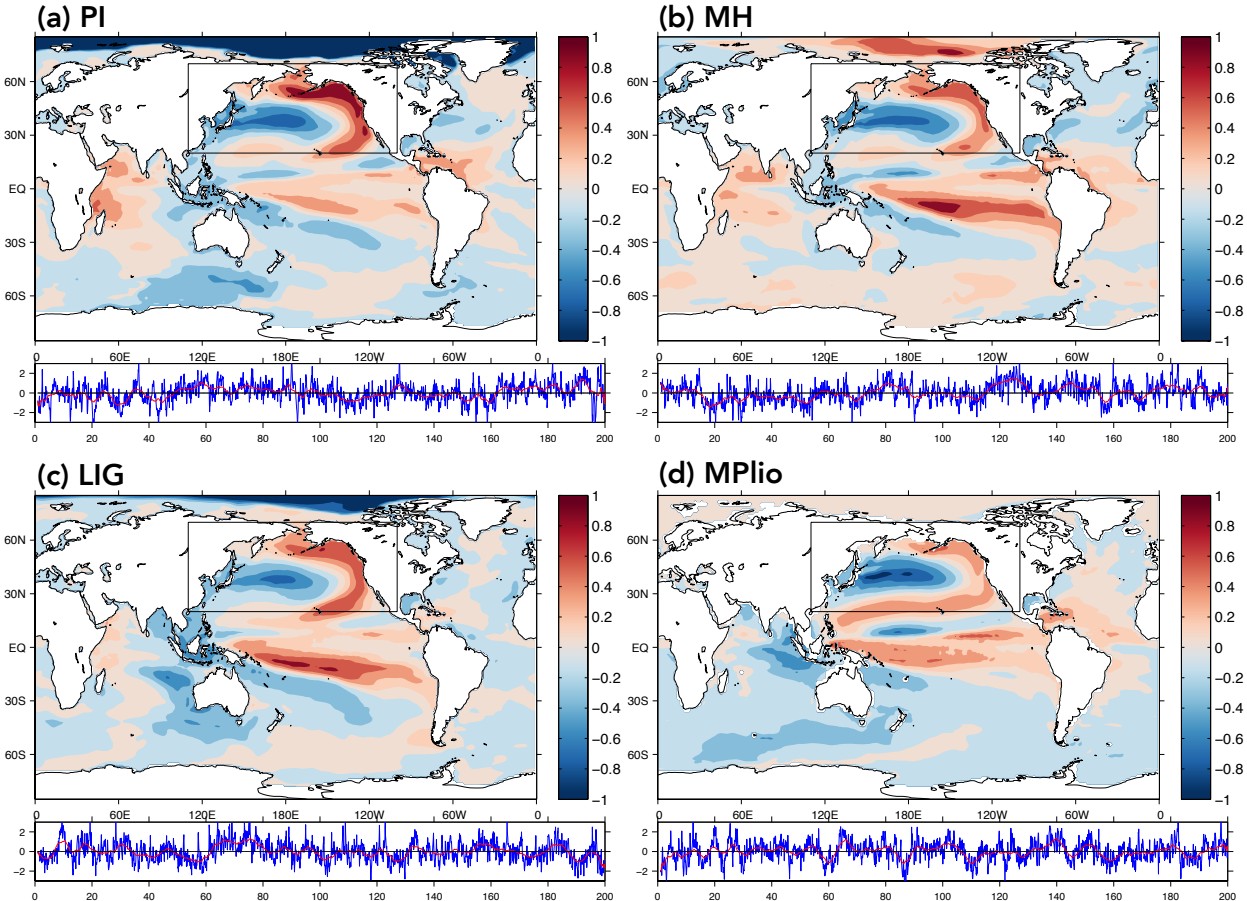

**Figure 15**. Simulated Pacific Decadal Oscillation (PDO) patterns. The spatial patterns show the leading empirical orthogonal function (EOF) of SST anomalies over the North Pacific (after removing the global mean SST anomaly). Note that the EOF calculation is restricted to the North Pacific (outlined by the black box). The global pattern is the regression coefficient of the monthly SST anomalies at each location onto the normalized principal component (PC) time series (unit: °C/°C). The time series shows the associated PC time series (blue line) and the 10-year running mean (red line).

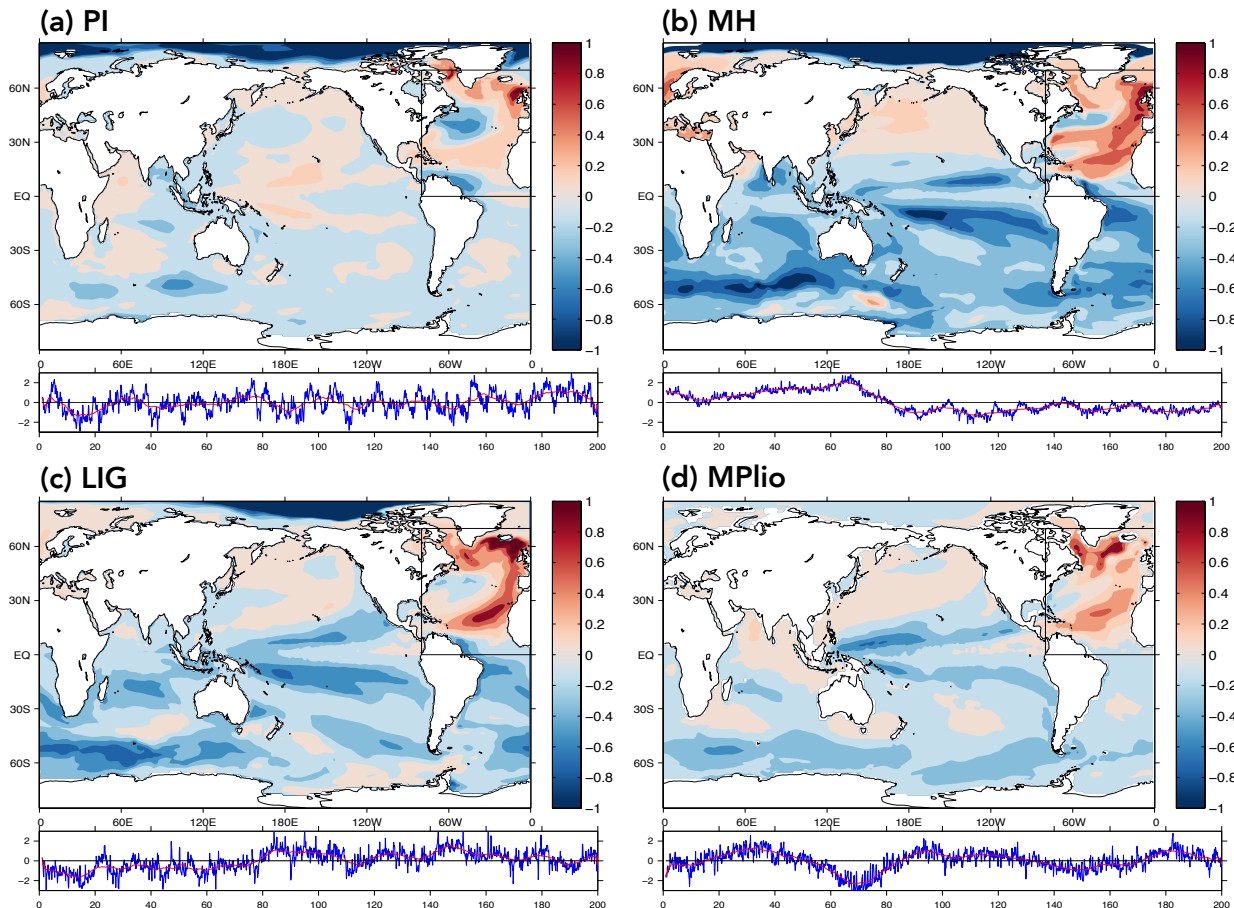

**Figure 16**. Simulated Atlantic Multidecadal Oscillation (AMO) patterns. The spatial patterns and time series are derived using the same EOF and regression method as PDO, but for the North Atlantic region (outlined by the black box).

The simulations show a decline of ENSO intensity from the *piControl* to the *midHolocene*, the *lig127k* and the *midPliocene*, which is universal across the equatorial central/eastern Pacific (Fig. 14). The Atlantic Niño variability is consistently weakened. The reduced ENSO variability during the Mid-Holocene is consistent with previous studies, either from proxy reconstructions (see a synthesis in Emile-Geay et al. (2016)) or model simulations (see a review by Lu et al. (2018)). The larger insolation anomalies in the *lig127k* compared to the *midHolocene* (Fig. 1) contribute to the more robust weakening of ENSO variability through reduced Bjerknes feedbacks (Liu et al., 2014;Lu et al., 2019).

The simulated PDO patterns generally reproduce the positive SST anomalies comparable in the equatorial eastern Pacific and the Northeast Pacific and negative anomalies in the central and western North Pacific (Fig. 15). This internal PDO variability

in paleoclimate conditions is consistent with a previous mid-Holocene study using earlier-generation climate models (An and Park, 2013). However, different simulations show remarkable changes in the relative amplitude: in the *midHolocene* and *lig127k,* the anomalies in the tropics are much larger and are almost equal to those in the Northeast Pacific. In the *midPliocene,* the overall anomalies are weaker and stretch westward to the west coast of the tropical Pacific, contributing to the westward shift of Walker circulation (Fig. 8f).

The AMO patterns in the EC-Earth3-LR simulations are characterized by a basin-wide mode in the North Atlantic (Fig. 16). The low-frequency mode has also been simulated in an earlier study during the Holocene (Wei and Lohmann, 2012). These anomalies also have a horseshoe shape (Gastineau and Frankignoul, 2015) with larger subpolar and subtropical anomalies. The spatial patterns of the AMO are mostly unchanged in the *piControl*, *lig127k* and *midPliocene* simulations but seem more predominant in the *midHolocene*.

The spectral feature of the simulated ENSO, PDO, and AMO are detected using the power spectrum method. The power spectra show that the dominant periodicities of ENSO, PDO and AMO lie in the 2-7 year, 8-20 year and 20-40 year time scales (Fig. S6), all are statistically significant, although the spectral peak differs slightly. These periodicities are consistent with present-day observation (see a review by Deser et al. (2010)) and highlight the model's capability to reproduce the representative natural internal variability.

**5 Conclusion**

This paper documents the four PMIP4-CMIP6 experiments performed by EC-Earth3-LR by providing the information on the model, experiments setup and results on large scale climate responses. The model outputs from the four simulations are published on ESGF for more applications by the research community. We expect that these data will be further used in studies through model-data comparison to understand the past climate change and climate variability, model-model and model-data comparison to improve the climate model performance, or multi-periods comparison to obtain an overview of the climate history.

We have followed PMIP4 protocol and adapted the climate forcing and boundary conditions in the EC-Earth3-LR. For the two interglacial simulations, most of the boundary conditions are prescribed as the pre-industrial conditions. The boundary conditions such as vegetations, aerosols, dust forcings, are essential to sensitivity studies and should be appropriately prescribed during these climate conditions in the future simulations. We try to perform the spin-up runs as long as possible, but due to limited computation resources and time constraints, the deep ocean may not be fully equilibrium in some of the simulations. The surface parameters have reached an acceptable equilibrium state as required by PMIP4.

The EC-Earth3-LR simulates reasonable climate response during past warm periods, as shown in paleo reconstructions and the other PMIP4-CMIP6 model ensemble. The different forcings cause these warm periods: the insolation change forces the two interglacial simulations, and the mid-Pliocene simulation is forced by the high $CO_2$ concentration comparable to today. The results show that the climate response in the latter forcing is dramatically more significant than the former ones, implying the ongoing global warming is critical in the context of long climate history.

Climate changes are amplified over the land in the higher latitudes, especially in the Arctic region. Arctic warming is apparent in all simulations with a strong link between the seasonal cycles of surface air temperature and sea ice and a strong link between the surface air temperature and insolation anomalies in the *midHolocene* and *lig127k* simulations. The climatic response in the Antarctic region is less prominent, and only small changes are observed in the two interglacial simulations.

The temperature gradient changes lead to a poleward expansion of Hadley circulation in all three warm period simulations, with reduced intensity in the Northern Hemisphere. In the Southern Hemisphere, the Hadley circulation strengthened in the *midHolocene* and *lig127k* but weakened in the *midPliocene*. We also found a strengthening and westward shift of the Pacific Walker circulation in all the three the three periods related to the ENSO changes.

The global monsoon area increased in the *midHolocene* compared to the *piControl,* with most of the increase located in the SH. However, the SH monsoon intensity decreased due to stronger seasonal precipitation differences. In *lig127k,* the global monsoon area expanded northward, and the monsoon intensity increased in the NH while it decreased in SH due to changes in the land-ocean thermal contrast in both hemispheres. For the *midPliocene*, both the global monsoon area and intensity increased, where the NH mainly contributed to the area increase and the SH to the intensity increase.

The EC-Earth3-LR reproduces reasonable spatial patterns and frequency features of several intrinsic climate variability modes such as ENSO, PDO and AMO. ENSO variability is weakened from pre-industrial towards *midHolocene*, *lig127k* and *midPliocene*, in response to intensified insolation changes in seasonality. The changes in PDO and AMO are more subtle and complex and will need further investigation.

-----------------------

*Code and data availability.* The code of EC-Earth3 is not publicly archived because of the copyright policy of the EC-Earth community. The model outputs for the four simulations performed and analyzed in this study are distributed and made freely available through the Earth System Grid Federation (ESGF).

The *piControl* (http://doi.org/10.22033/ESGF/CMIP6.4847);

The *midHolocene* (http://doi.org/10.22033/ESGF/CMIP6.4801);

The lig127k (http://doi.org/10.22033/ESGF/CMIP6.4798);

The *midPliocene* (http://doi.org/10.22033/ESGF/CMIP6.4804).

Details on the ESGF can be found on the website of the CMIP Panel (https://www.wcrp-climate.org/wgcm-cmip/wgcm-cmip6).

The reconstruction data used for data-model comparison is in the Supplement.

*Supplement.* The supplement related to this article is attached as *PMIP4-EC-Earth-supplement.doc*, and Supplementary_Model-data_comparison.xlsx.

*Author contributions.* Qiong Z. was responsible for the development and description of EC-Earth3-LR for PMIP4-CMIP6, supervised the modelling and the analyses. Qiang Z. executed the *piContol*, the *midHolocene* and the *lig127k* experiments. QL setup and run the *midPliocene* experiment. ZH analyzed global circulation analyses. WN analyzed polar climate response. EB, JA and JC analyzed the global monsoon, and ZL analyzed climate variability modes. KW and SY contributed to the preparation of initial conditions and commented on the description of EC-Earth3.

*Acknowledgements.* This research has been supported by the Swedish Research Council (Vetenskapsrådet, grant no. 2013-06476 and 2017-04232). The model simulations with EC-Earth3 and data analysis were performed using ECMWF's computing and archive facilities and the Swedish National Infrastructure for Computing (SNIC) at the National Supercomputer Centre (NSC) partially funded by the Swedish Research Council through grant agreement no. 2016-07213. HadISST data were obtained from https://www.metoffice.gov.uk/hadobs/hadisst/ and are © British Crown Copyright, Met Office, 2020, provided under a Non-Commercial Government Licence http://www.nationalarchives.gov.uk/doc/non-commercial-government-licence/version/2/

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
