# Peer review of "Simulating the mid-Holocene, Last Interglacial and mid-Pliocene climate with EC-Earth3-LR"

_Geoscientific Model Development, 2020_

## Referee Comment (RC1) · Anonymous Referee #1 · 7 Oct 2020

In the paper, Zhang et al document the PMIP4 experiments run with EC-Earth3-LR, and further illustrate the simulated global and regional climate responses. The paper is well written. I have one suggestion for improving the paper. More information about the model, EC-Earth3, and its ability in simulating PI climate should be added in the revised version.

Line 84. Since the authors choose GMD to publish their PMIP experiments, I suggest that the authors add more information to introduce their model, rather than a simple sentence. "A detailed description of the ECEarth3 and its contribution to CMIP6 is documented in Döscher et al. (2020)."

Line 85-93, the resolution of other EC-Earth3 CMIP6 versions should be introduced here.

[Figure]

In the section 3.4, I suggest that some figures from Fig S1 (for PI simulations) should be included in the main texts, since these figures are important and helpful to show the modelling ability of EC-Earth3-LR.

Line 242, does the 0.5 W m-2 energy leak appear in PI control experiments run with other EC-Earth3 versions? How long is the spin-up of EC-Earth3-LR PI control run? This important information should be introduced here.

I suggest the authors reorganizing the section 3.4, and providing more information about their PI control experiment, together with some comparisons to other EC-Earth3 versions.

---

## Referee Comment (RC2) · Anonymous Referee #2 · 12 Oct 2020

This paper presents the results of the EC-Earth3-LR for simulating the three past warm periods (mid-Holocene, Last Interglacial, and mid-Pliocene) which are documented in the Paleoclimate Model Intercomparison Project (PMIP) phase 4. In this paper, Zhang et al provide a comprehensive and diagnostical analysis on the modeling results. This work is very meaningful and valuable for the PMIP4 group. I would recommend its publication after addressing the comments as follows :

1. The introduction is a bit broadly. I suggest the authors can provide more information on the three past warm periods rather than giving a general introduction for all PMIP4 target periods. 2. In section 2.2.2, the authors organize a whole paragraph to describe the new albedo parameterization of snow on ice-sheet, but for the readers, we still have no idea about the core information of this new scheme. I suggest the authors
can put forward directly the core information about this scheme rather than citing a reference. 3. I suggest to change the title of section 4.1, since the authors present both SAT and PRECIP anomalies. Or the authors can integrate section 4.1 and 4.2 as one section describing the response of global mean temperature and precipitation. 4. In line 333, "and induce weaker Hadley circulation (Fig. 5c)", is it weaker or stronger ? 5. In Figure 5, could the authors provide the contour interval value. For the walker circulation, the authors interpret westward shifts of the ascending part in lig127k and midpliocene. For the readers, it is not straightforwardly understood in Fig 5. Could the author provide a supplement fig for the climatology result of these two simulations for the walker circulation ?

6. How do the authors define the sea ice edge in Fig 7 ? The Arctic minimum SIE in midpliocene in Fig 7 disappears, does it mean no sea ice in midpliocene in Aug ? However, in Fig 6, the Arctic SIE anomalies between midpliocene and piControl show that the SIE anomalies in Auguste is weaker than that in March, which seems not consistent with the sea ice edge presented in Fig 7.

7. Last but very important question, how do the model results comparing with the records ? Since the authors highlight "the ability of the model to capture the climate response under different climate forcings, providing potential implications for confidence in future projections with EC-Earth model"in their abstract. It is better to examine the model performance with the related reconstructions.

---

## Author Comment (AC1) · 30 Nov 2020

Thanks for your comments, follow your suggestions we have provided more information and revised the manuscript, below is our response to your comments.

1. Indeed, we should provide more information on the model since the reference paper is still in preparing. In the revised version we have added more information on different model component.

2. We have now introduced the model resolution used in the other EC-Earth3 CMIP6. The standard CMIP6 model version with atmosphere resolution T255L91 (horizontal $\sim$ 80 km) and ocean resolution ORCA1L75 (horizontal 1x1) is used in most of the MIPs (e.g., CMIP, DCPP, LS3MIP, PAMIP, RFMIP, ScenarioMIP, VolMIP,

CORDEX, DynVarMIP, SIMIP). The high-resolution model version with atmosphere resolution T511L91 (horizontal ∼ 25km) and ocean resolution ORCA025L75 (horizontal 0.25x0.25), is used in the decadal prediction DCPP and HighResMIP. These are included in the revised version.

3. Fig S1 is to show the spinning up process and we prefer to keep that in the supplement. We agree that an evaluation for simulated PI climate is necessary. Beside showing the comparison of the simulated PI temperature with reanalysis, we now added a new figure on comparison of the simulated PI precipitation with ERA20C. The comparisons show that the EC-Earth3 model does have the cold and dry biases in northern Hemisphere and warm and wet biases in southern Hemisphere in PI control.

4. The imblance between net TOA and net surface radiative fluxes is different with different resolution. In the EC-Earth3 standard resolution T255L91, the difference is in the order of 0.25 W/m2 (Döscher et al., 2020) and for the high resolution T511L91 the imbalance is 0.9 W/m2 (personal communication). We have revised the text to provide the citable information for T255L91.

We now improved the section 3.4 as suggested. The spin-up of EC-Earth3-LR PI control is run for approximately 1000 years during the course of development and tunning process. The 200 years spin-up run shown in FigS1 is continued from the previous long spin-up by applying the implemented physics described in section 2. In the revision we have added more information on the PI run, as well as the comparison to PI runs with other model versions.

---

## Author Comment (AC2) · 30 Nov 2020

Thanks for the review comments, we have taken into account the suggestions and revised the manuscript accordingly. Below is our response to your comments.

1. As suggested, we have added more motivation and description on the three past warm periods in the introduction. We also included summary on simulated large-scale features by the PMIP4 models on mid-Holocene (Brierly, et al., 2020), lig127K (Otto-Bliesner et al., 2020) and mid-Pliocene (Haywood et al., 2020).

2. The core information for introducing a new albedo parameterization is to better represent the albedo-feedback by allowing the snow can fall and melt over the icesheet, instead of static in previous scheme. We now mention this core information in the

beginning of section 2.2.2.

3. Thanks, as suggested we now combine 4.1 and 4.2 into one section to describe the global features of temperature and precipitation. We also added a global change in precipitation.

4. Thanks for your carefully reading and observed this mistake. Indeed, it should be "stronger" Hadley circulation but not the "weaker" as stated in the manuscript. As shown in Figure 4b and 4c, the midHolocene and lig127k simulations show slight cooling in the SH compared to the tropics. This can induce a slightly stronger meridional temperature gradient and favor a stronger Hadley circulation in Figure 5a and c. This is corrected in the revised version.

5. In Figure 5, we have compared the PI climatology of vertical integrated zonal mass stream function (ZMS) (in contours) and the anomalous ZMS in all three warm periods (shading), they all exhibit positive anomalies west of the positive Centre in PI run, which is indicating the westward movement of the Walker circulation. The westward shift appears more evident in the lig127k and midPliocene than in the midHolocene simulation. We now provided the climatology for all the simulations in the supplement for a better comparison.

6. The sea ice extent defines a region as either "ice-covered" or "not ice-covered." In the model grid, for each grid cell, a threshold determines either the cell has ice or the cell has no ice. Here we apply a commonly used threshold 15% (such as used by National Snow and Ice Data Center NSIDC), meaning that if the model grid cell has greater than 15% ice concentration, the cell is "ice-covered." A threshold can also be as high as 30 percent. The sea ice edge is the 15% sea ice concentration isocline. We added our definition of sea ice and sea ice edge in the text.

Indeed, the mPlio-PI anomaly is smaller in August than in March, simply because the difference between the two simulations is smaller then. In August, the PI simulation has less than $10*10^6$ km$^2$ of sea ice, while the mPlio simulation has 0. Therefore,

the anomaly is smaller than 10*10ˆ6 kmˆ2 in this month. On the contrary, in March the PI simulation has more sea ice (winter peak) and also a larger difference with the mPlio simulation, of almost 12*10ˆ6 kmˆ2 more sea ice. Therefore, Figure 6 and 7 are consistent with each other."

7. Thanks for the comments. We collected the published SST reconstructions for all three warm periods and added a data-model comparison on SST in section 4.2.
* * *

---

## Author Response (AR1)

In the paper, Zhang et al document the PMIP4 experiments run with EC-Earth3-LR, and further illustrate the simulated global and regional climate responses. The paper is well written. I have one suggestion for improving the paper. More information about the model, EC-Earth3, and its ability in simulating PI climate should be added in the revised version.

Line 84. Since the authors choose GMD to publish their PMIP experiments, I suggest that the authors add more information to introduce their model, rather than a simple sentence. "A detailed description of the ECEarth3 and its contribution to CMIP6 is documented in Döscher et al. (2020)."

Thanks for the comment, in the revised version we have added more information on different model component in section 2.1, Page 3, L70-101.

Line 85-93, the resolution of other EC-Earth3 CMIP6 versions should be introduced here.

We have now introduced the model resolution used in the other EC-Earth3 CMIP6 in section 2.1, Page 3, L87-94.

It reads as:

EC-Earth3 earth system model version also includes the atmospheric chemistry component TM5, the dynamic vegetation model LPJ-Guess and the ocean biogeochemistry component PISCES. The EC-Earth3 model contributes to CMIP6 in several configurations by using different resolutions or including different components. A detailed description of the EC-Earth3 and its contribution to CMIP6 is documented in Döscher et al. (2020). For example, regarding the resolution, the standard version EC-Earth3 with atmospheric resolution T255L91 (horizontal ~ 80 km) and ocean resolution ORCA1L75 (horizontal 1°) is used in most MIPs (e.g., CMIP, DCPP, LS3MIP, PAMIP, RFMIP, ScenarioMIP, VolMIP, CORDEX, DynVarMIP, SIMIP). The high-resolution model version with atmospheric resolution T511L91 (horizontal ~ 25km) and ocean resolution ORCA025L75 (horizontal 0.25°) is used for DCPP and HighResMIP decadal predictions.

The PMIP4 experiments require long spin-up and equilibrium simulations which are computationally demanding. We thus use model configurations without atmospheric chemistry and ocean biogeochemistry at low resolution, with atmospheric resolution T159L62 (horizontal ~ 125km) and ocean resolution ORCA1L75. For the *piControl*, the *midHolocene*, the *lig127k*, *lgm* and the *midPliocene*, we have used the EC-Earth3-LR, which is a configuration of EC-Earth3 with only atmosphere, ocean and sea ice components at Low-Resolution (LR). The coupled vegetation LPJ-GUESS is used for the *past1000* simulation with model configuration named EC-Earth3-veg-LR.

Despite using the lowest resolution among the EC-Earth3 CMIP6 MIPs configuration, EC-Earth3-LR has a relatively high resolution compared to the other PMIP4 models (Brierley et al., 2020).

In the section 3.4, I suggest that some figures from Fig S1 (for PI simulations) should be included in the main texts, since these figures are important and helpful to show the modelling ability of EC-Earth3-LR.

We now moved energy balance figure S1 to main text as Figure 3 and keep the other three in the supplement. For the evaluation of simulated PI climate, besides the comparison of the simulated PI temperature with reanalysis, we now added a new figure on comparison of the simulated PI precipitation with ERA20C as Figure 6. The comparisons show that the EC-Earth3 model does have the cold and dry biases in northern Hemisphere and warm and wet biases in southern Hemisphere in PI control.

New Fig.3 can be found Page 10, related text is in Page 12 L237-256.

New Fig.6 can be found in Page 16, related text is in Page 17 L325-330.

Line 242, does the 0.5 W m-2 energy leak appear in PI control experiments run with other EC-Earth3 versions? How long is the spin-up of EC-Earth3-LR PI control run? This important information should be introduced here.

I suggest the authors reorganizing the section 3.4, and providing more information about their PI control experiment, together with some comparisons to other EC-Earth3 versions.

The energy imbalance differs for different resolution tunning. For example, in the EC-Earth3 standard resolution T255L91, the energy imbalance is in the order of 0.25 W m-2 (Döscher et al., 2020), and in the high resolution T511L91 tunning version the imbalance is 0.9 W m-2 (personal communication). We have revised the text to provide the information of energy imbalance for T255L91 and T511L91 on Page 12 L241-243.

The spin-up of EC-Earth3-LR PI control is run for approximately 1000 years during the course of development and tunning process. We have run another 200 years. The 200 years spin-up run shown in Fig.S1(now Fig.3) is continued from the previous long spin-up by using the implemented physics. This is mentioned in the beginning of Sect. 3.4 on Page 11 L225-235.

**Anonymous Referee #2**

This paper presents the results of the EC-Earth3-LR for simulating the three past warm periods (mid-Holocene, Last Interglacial, and mid-Pliocene) which are documented in the Paleoclimate Model Intercomparison Project (PMIP) phase 4. In this paper, Zhang et al provide a comprehensive and diagnostical analysis on the modeling results. This work is very meaningful and valuable for the PMIP4 group. I would recommend its publication after addressing the comments as follows :

1. The introduction is a bit broadly. I suggest the authors can provide more information on the three past warm periods rather than giving a general introduction for all PMIP4 target periods.

As suggested, we removed the mention of other PMIP4 target periods in the introduction, and wrote more on the motivation for simulating the past warm periods.

2. In section 2.2.2, the authors organize a whole paragraph to describe the new albedo parameterization of snow on ice-sheet, but for the readers, we still have no idea about the core information of this new

scheme. I suggest the authors can put forward directly the core information about this scheme rather than citing a reference.

The core information for introducing a new albedo parameterization is to better represent the albedo-feedback by allowing the snow can fall and melt over the icesheet, instead of static in previous scheme. We now mention this core information in the beginning of section 2.2.2 in Page 4 L122-125.

3. I suggest to change the title of section 4.1, since the authors present both SAT and PRECIP anomalies. Or the authors can integrate section 4.1 and 4.2 as one section describing the response of global mean temperature and precipitation.

Thanks, as suggested we now combine 4.1 and 4.2 into one section 4.1 to describe the large-scale features of climate change. We added a figure for global change in precipitation as Figure 6. A data-model comparison on SST is presented in the following section 4.2. Now the global changes in three commonly used variables (surface air temperature, precipitation and SST) are presented in Fig 4-6.

4. In line 333, "and induce weaker Hadley circulation (Fig. 5c)", is it weaker or stronger ?

Thanks for your carefully reading and observed this mistake. Indeed, it should be "stronger" Hadley circulation but not the "weaker" as stated in the manuscript. As shown in Figure 4b and 4c, the *midHolocene* and *lig127k* simulations show slight cooling in the SH compared to the tropics. This can induce a slightly stronger meridional temperature gradient and favor a stronger Hadley circulation in Figure 5a and c. This is corrected in Page 20 L395.

5. In Figure 5, could the authors provide the contour interval value. For the walker circulation, the authors interpret westward shifts of the ascending part in lig127k and midpliocene. For the readers, it is not straightfordly understood in Fig 5. Could the author provide a supplement fig for the climatology result of these two simulations for the walker circulation ?

Label all the contour interval in Fig 5 (now Fig 8) looks messy. To show the shifts more clearly, we present the climatology of vertical integrated zonal mass stream function (ZMS) of three simulations in Fig 10. Compared to the *piControl*, the most striking features in all three warm period simulations are westward movement of the Walker circulation. Especially, there is more westward shift in the *lig127k* and *midPliocene* with respective to *midHolocene* simulation. There features are consistent with the results in Figure 5.

New Fig 9 can be found in Page 20, related text is in Page 20 L395-399.

6. How do the authors define the sea ice edge in Fig 7 ? The Arctic minimum SIE in midpliocene in Fig 7 disappears, does it mean no sea ice in midpliocene in Aug ? However, in Fig 6, the Arctic SIE anomalies between midpliocene and piControl show that the SIE anomalies in August is weaker than that in March, which seems not consistent with the sea ice edge presented in Fig 7.

The sea ice extent defines a region as either "ice-covered" or "not ice-covered." In the model grid, for each grid cell, either the cell has ice (usually a value of "1") or the cell has no ice (usually a value of "0"). A threshold determines this labeling. Here we apply a commonly used threshold 15% (such as used by National Snow and Ice Data Center NSIDC), meaning that if the model grid cell has greater than 15% ice concentration, the cell is labeled as "ice-covered." A threshold can also be as high as 30 percent. The sea ice edge is the 15% sea ice concentration isocline. We added our definition of sea ice in the text on Page 21 L450-454.

Indeed, the sea ice in the mid-Pliocene disappears in August and that is why there is no sea ice edge in Figure 7c (we showed for September instead of August) for the mid-Pliocene. We mentioned this in Page 22 L479-48 as "midPliocene SIE anomalies are lowest in the summer because low sea ice

concentrations during this season limit SIE decrease in the mid-Pliocene, while SIE can still decrease more in the PI simulation." It is true that the SIE anomaly is smaller in melting season than in frozen season. This is because during mid-Pliocene all sea-ice becomes seasonal sea-ice, and during PI there are perennial sea-ice However, in August the PI SIE is also smaller, and since the SIE in the mid-Pliocene is already small in July and then zero in August (little decrease), the anomaly is smaller.

7. Last but very important question, how do the model results comparing with the records ? Since the authors highlight "the ability of the model to capture the climate response under different climate forcings, providing potential implications for confidence in future projections with EC-Earth model"in their abstract. It is better to examine the model performance with the related reconstructions.

Thanks for the comments. We use the available SST reconstructions for lig127k and mid-Pliocene, add a data-model comparison on SST as Fig 4.6 in section 4.2, Page 17-18, L334-377.

---

## Author Response (AR2)

Dear Editor,

Thanks for your careful reading, below is our response to your comments in blue font.

Comments to the Author:

Line 84 states that "the time step for IFS is 60 minutes". Is it the full time step of IFS, i.e. the time step at which all processes are solved, including the radiation scheme, the soil model, river routing, etc... ?
Same question for NEMO. 45 minutes is the time step generally used in ORCA1L75 for the dynamics (OPA). The ice model (LIM) has generally a longer time step, often synchronised with the coupling ones. That means a 3h time step for NEMO as a whole, with sub time stepping for the ocean dynamics. Please clarify this for both models.

The time step of IFS at T159L62 resolution is 60 minutes except for radiation that is solved only every 3 hours (but updated with cloud cover information in between two full computations). The time step for NEMO at ORCA1L75 resolution is 45 minutes, we use the same time step for the ocean and for the sea-ice.

This is clarified in the text L84-86.

Line 84: "for NEMO is 45 minutes" -> "for NEMO it is 45 minutes"

Revised as mentioned above.

SST, DJF, MAM, JJA, SON, MJJAS and NDJFM should be explicated as the first appearance.
Add a white space after semicolon in the lists of citations.

As suggested, all these abbreviations are explicated at the first mention.

Remove a few double spaces remaining in the text.

Checked and removed a few double spaces, some visual large space is due to the "Justify Text" format in Microsoft word, should disappear in the final publication.

Line 110: "two types of the year" -> "two type of year" (??)

Corrected.

Line 149: W/m2 -> W m2 to be consistent with the rest of the text

Corrected.

Line 437: (Hind et al., 2016) -> Hind et al.(2016)

Corrected.

Fig.9: Meridioanl -> Meridional / Zoanl -> Zonal

Corrected.

Fig. 10: bottom left panel 10^6km^2 -> 10^6 km^2 (add a space)

Revised.